# Steady state evoked potential (SSEP) responses in the primary and secondary somatosensory cortices of anesthetized cats: Nonlinearity characterized by harmonic and intermodulation frequencies

Yota Kawashima[1], Rannee Li[1], Spencer Chin-Yu Chen[2], Richard Martin Vickery[3], John W. Morley[4], Naotsugu Tsuchiya[1,5,6]*

**1** Turner Institute for Brain and Mental Health, School of Psychological Science, Monash University, Melbourne, Victoria, Australia, **2** Department of Neurosurgery, Robert Wood Johnson Medical School, Rutgers University, New Brunswick, New Jersey, United States of America, **3** School of Medical Sciences, UNSW Sydney, Sydney, New South Wales, Australia, **4** School of Medicine, Western Sydney University, Penrith, New South Wales, Australia, **5** Center for Information and Neural Networks (CiNet), National Institute of Information and Communications Technology (NICT), Suita, Osaka, Japan, **6** Advanced Telecommunications Research Computational Neuroscience Laboratories, Soraku-gun, Kyoto, Japan

☯ These authors contributed equally to this work.
* naotsugu.tsuchiya@monash.edu

## Abstract

When presented with an oscillatory sensory input at a particular frequency, $F$ [Hz], neural systems respond with the corresponding frequency, $f$ [Hz], and its multiples. When the input includes two frequencies ($F1$ and $F2$) and they are nonlinearly integrated in the system, responses at intermodulation frequencies (i.e., $n1 * f1 + n2 * f2$ [Hz], where $n1$ and $n2$ are non-zero integers) emerge. Utilizing these properties, the steady state evoked potential (SSEP) paradigm allows us to characterize linear and nonlinear neural computation performed in cortical neurocircuitry. Here, we analyzed the steady state evoked local field potentials (LFPs) recorded from the primary (S1) and secondary (S2) somatosensory cortex of anesthetized cats (maintained with alfaxalone) while we presented slow ($F1$ = 23Hz) and fast ($F2$ = 200Hz) somatosensory vibration to the contralateral paw pads and digits. Over 9 experimental sessions, we recorded LFPs from $N$ = 1620 and $N$ = 1008 bipolar-referenced sites in S1 and S2 using electrode arrays. Power spectral analyses revealed strong responses at 1) the fundamental ($f1$, $f2$), 2) its harmonic, 3) the intermodulation frequencies, and 4) broadband frequencies (50-150Hz). To compare the computational architecture in S1 and S2, we employed simple computational modeling. Our modeling results necessitate nonlinear computation to explain SSEP in S2 more than S1. Combined with our current analysis of LFPs, our paradigm offers a rare opportunity to constrain the computational architecture of hierarchical organization of S1 and S2 and to reveal how a large-scale SSEP can emerge from local neural population activities.

**Data Availability Statement:** All data file and analysis scripts are available from OSF database (DOI: 10.17605/OSF.IO/CQRFJ).

**Funding:** This study was supported by the Australian Research Council to NT (FT120100619, DP130100194, DP180104128, and DP180100396) and the Australian Research Council Thinking Systems to JM (TS0669860).

**Competing interests:** The authors have declared that no competing interests exist.

## Introduction

Internal processing architecture of physical systems, including neuronal networks of the brain, can be characterized by probing them with steady oscillatory input at a particular frequency and measuring their output in the frequency domain [1–4]. In sensory neuroscience, this technique is called a "steady state evoked potential (SSEP)" paradigm. With the oscillatory input probe, the system's output is examined at various response frequencies. Throughout this paper, we denote input and output frequency with a capital, $F$ [Hz], and a small letter, $f$ [Hz], respectively. A linear-time-invariant (LTI) system, which does not contain any nonlinearity in the system, can only modulate the amplitude or phase of the responses at $f = F$ [Hz] and cannot generate responses that are not present in the stimulus [4]. The presence of neural response at the harmonics of the input frequency ($n*f$, where $n = 2, 3, 4, \ldots$) necessitates nonlinear processing in the system. Recently, the SSEP paradigm and its variants have arisen as a powerful technique in sensory neuroscience to characterize properties of the sensory neural circuits in the visual, auditory and somatosensory systems in various animal species [3–8].

The SSEP paradigm is particularly powerful when extended to use multiple input frequencies. For the case of two input frequencies ($F1$, $F2$), as used in this paper, in addition to the harmonic frequencies, nonlinear processing in the system can be further inferred by the presence of intermodulation frequencies ($n1*f1+n2*f2$) in the output of the system [1, 2, 6, 9–14] (See a review [15]). Intermodulation responses can only arise from nonlinear processing of two (or more) input frequencies, e.g., $F1$ and $F2$. Recent cognitive neuroscience investigations have utilized this theoretical prediction [4–6, 11–13]. While these intermodulation phenomena are attracting popularity in large scale EEG measures in human cognitive neurosciences, the detailed neuronal mechanisms on how intermodulation arises at the level of single neurons and local neural circuitry remains unclear.

Previously, we applied the SSEP paradigm in the somatosensory domain in the anesthetized cats [16], focusing on the analysis of isolated single (or multiple) neuron spiking activities. There, we observed neurons responding in synchrony with the low frequency stimulus ($F1 = 23$Hz), and neurons that seemed partially time-locked to the high frequency input ($F2 = 200$Hz) (data not published). Further we found some neurons that responded to low ($F1 = 23$Hz) and high ($F2 = 200$Hz) frequency vibrations in a manner where the spiking rates were linearly related to the amplitude of the respective vibratory inputs. We also found that some neurons responded to these vibratory inputs in a supralinear facilitatory manner.

In this paper, we turned our analysis to local field potentials (LFPs) recorded in the same experiments in order to examine the population-level neuronal responses at the harmonic and intermodulation frequencies of the input stimuli. While a single cortical neuron may not be able to respond to every cycle of the very high frequency stimulus (e.g., $F2 = 200$Hz) due to the refractory period, neurons are capable of collectively encoding every cycle at such a high frequency. As LFPs reflect collective actions of neurons, we examine LFPs for high frequency neural population response and evidence of nonlinear processing that manifests as responses at harmonic ($2f1$, $3f1$, etc) and intermodulation ($n1*f1+n2*f2$) frequencies.

We found that LFPs recorded in the primary and secondary somatosensory cortices of anesthetized cats indeed show strong evidence of nonlinear processing from the examination of the SSEP harmonic and intermodulation. Unexpectedly, we also found that non-harmonic and non-intermodulation frequencies between 50-150Hz in the secondary somatosensory cortex can be strongly modulated by the strength of the vibratory inputs. Finally, our computational modeling suggests that nonlinear processing is prominent in the secondary somatosensory cortex, but less so in the primary sensory cortex. Our results constrain the network architecture of

the somatosensory cortical circuits. It will inform future studies that aim to generalize our findings to other sensory modalities, animal models, and conscious states.

## Methods

### Experimental subjects and procedures

The detailed experimental methods are described in [16]. Here we describe the aspects of the protocol that are relevant for our current paper. This study was carried out in strict accordance with the recommendations in the Guide for the Care and Use of Laboratory Animals of the National Health and Medical Research Council, Australia. All procedures involving animals were approved and monitored by the University of New South Wales Animal Care and Ethics Committee (project number: ACEC 09/7B). Animals were sourced from a licensed supplier as per Animal Research Act 1985 and housed in the same facility under care of certified veterinarian. They were group housed and were free to roam within the enclosed space. All animals were on standard diet, water, environmental enrichment, and day/night cycle. All surgery was performed under ketamine-xylazine anesthesia, and all efforts were made to minimize suffering.

Outbred domestic cats had anesthesia induced with an intramuscular dose of ketamine (20 mg/kg) and xylazine (2.0 mg/kg). Anesthesia was maintained over the three days of an experiment by intravenous infusion of alfaxalone (1.2 mg/kg) delivered in an equal mixture of Hartmann's solution and 5% glucose solution at approximately 2 ml/kg/hr. The animal received daily doses of dexamethasone (1.5 mg/kg) and a broad spectrum antibiotic (Baytril, 0.1 ml/kg) intramuscularly, and atropine (0.2 mg/kg) subcutaneously.

The animal was secured in a stereotaxic frame and a craniotomy and durotomy were performed to expose the primary and secondary somatosensory areas (Fig 1A). The exposed cortex was mapped by recording evoked potentials using a multichannel recording system (RZ2 TDT, Tucker Davis Technologies Inc., Florida, U.S.A) and an amplifier and headstage (model 1800, AMSystems, Washington, U.S.A.). At the end of the experiment, animals are euthanized with an overdose of pentobarbital. Lack of heart determined from ECG and from palpation was used to confirm euthanasia.

We used the RZ2 TDT system to drive a Gearing & Watson mechanical stimulator with a 5mm diameter flat perspex tip. As vibrotactile stimuli, we delivered *F1* = 23Hz or *F2* = 200Hz sinusoidal indentation to the paw pad or digit of the cat. We chose 23Hz and 200Hz to stimulate rapidly-adapting sensory endings (RAI) and very fast-adapting, so-called Pacinian (PC or RAII) respectively. The peak-to-peak amplitude of the sinusoid for 23Hz ranged from 0 to 159μm, while that for 200Hz ranged from 0 to 31μm. The probe at rest barely indented the skin. Hair around the forelimb paw pads was shaved to prevent activation during stimulation.

We recorded neural activity with a 10x10 "planar" multi-electrode array (Blackrock Microsystems, Utah, U.S.A) in S1 and with a 8x8 "linear" multi-electrode array (NeuroNexus, Michigan, U.S.A.) in S2. We inserted these arrays aiming at recording from the paw representation regions within S1 and S2. Through the mapping procedure, we confirmed these recording sites. We used the planar arrays to obtain wide coverage of S1 and nearby regions. The array consisted of 1.5mm long electrodes and recorded data from those electrodes across a 13mm$^2$ horizontal plane of cortex. We chose to use the linear array to record neural activity from S2 because S2 is buried in the depth of the cortex. The electrode contacts on the linear array recorded data from a vertical cross-section of multiple cortical layers along 1.4mm of cortex. Data from these arrays were collected using the RZ2 TDT multichannel recording system through a PZ2 TDT pre-amplifier. Streaming data were recorded simultaneously without filtering at 12kHz.

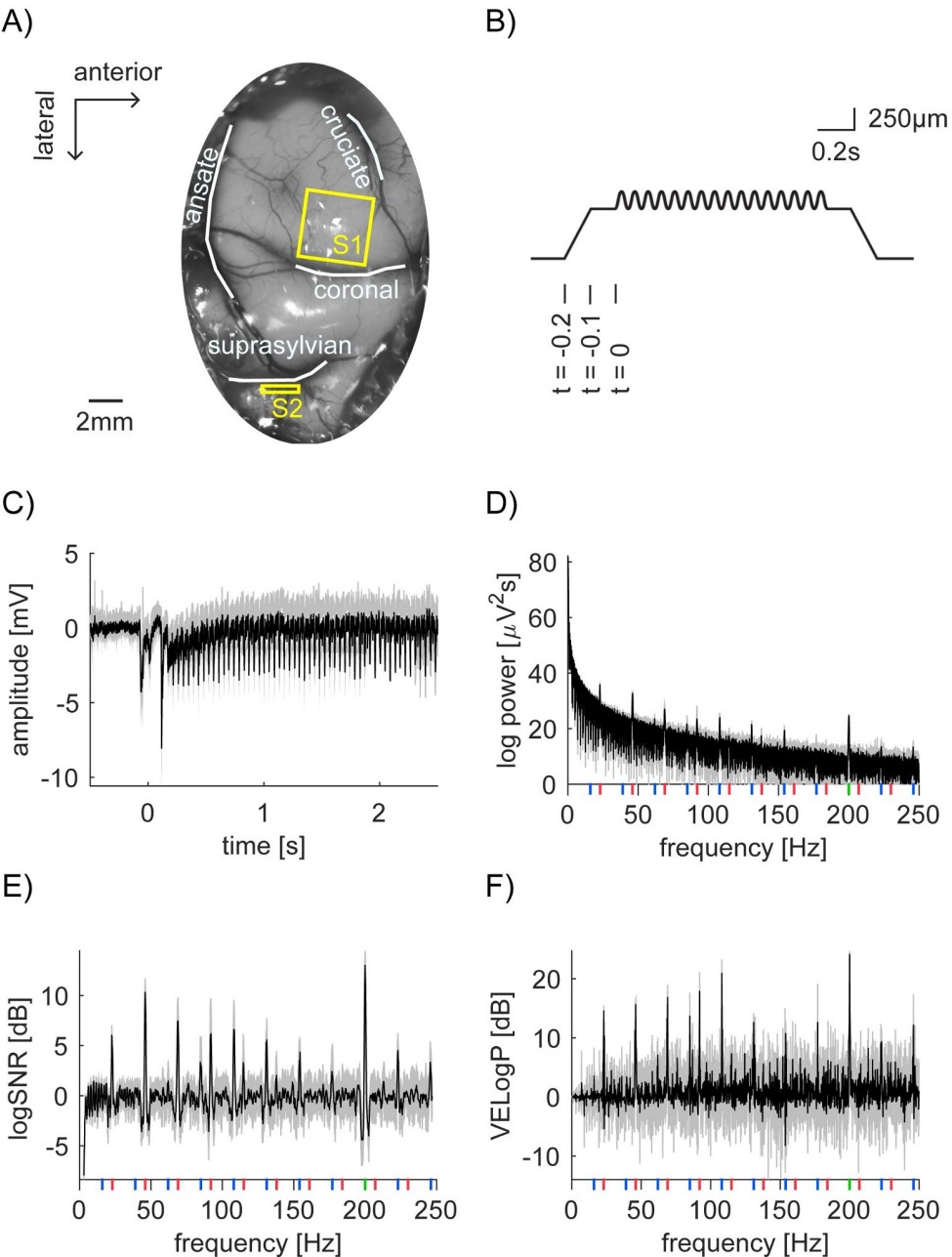

**Fig 1. Neural recording, vibratory stimulation and our analysis scheme.** (A) Photo of anterior parietal cortex with outlines of sulci (white lines) superimposed. The planar array was inserted into the paw representation region of S1 (yellow square). A linear array was inserted into S2 region located in the suprasylvian sulci (yellow rectangle). (B) Time course of the depth modulation of the vibratory stimulation. The stimulator presses 500μm into the skin over 0.1s of ramping, followed by a pause of 0.1s. At $t = 0$, the vibration stimulation starts. Here, 159μm of 23Hz sinusoid is superimposed on a step indentation. We represent a combination of the depth modulation for $F1$ and $F2$, such as $X$ [μm] and $Y$ [μm], as [$F1$, $F2$] = [$X$, $Y$]. (C) Time domain representation of LFP signal from bipolar channel 156 in S1 (Session 2–2). Here, we computed the mean of the pre-stimulus LFP from -0.5 to -0.1s and used it as a baseline per trial, which is subtracted from LFP per trial. The mean trace from 15 trials is shown ([$F1$, $F2$] = [159, 16]). Shade represents the standard deviation across 15 trials, indicating extremely robust and clean SSEP. (D) Power spectrum of the LFP signal from 0.5 to 2.5s after stimulus onset in (C). Again, shading represents standard deviation. Vertical lines show frequencies of interest. (Red for 23Hz and its harmonics, green for 200Hz and blue for intermodulation.) (E and F) Frequency domain representation of log$SNR$ (E) and vibration evoked logPower (VELogP) (F). Note that raw log values are multiplied by 10 and values are shown in [dB].

Stimulus duration varied across sessions (Table 1), ranging from 3 to 4.4s. The peak-to-peak amplitude of the low frequency sinusoid varied from 0 and 159µm, and the high frequency sinusoid from 0 to 31µm. The amplitudes for the two sinusoids were selected pseudor-andomly for each trial within each session. The number of trials per stimulus condition (a particular combination of *F1* and *F2* amplitude) ranged from 10–15 depending on the recording session.

## Data preprocessing

To reduce line noise and obtain finer spatial resolution, we first applied bipolar re-referencing to the original LFP data, by subtracting the unipolar referenced voltage of each electrode from the horizontal or vertical neighbouring electrode [17]. Throughout this paper, we call the resulting bipolar re-referenced data as "bipolar channels". Bipolar re-referencing resulted in 180 bipolar channels (90 vertical and 90 horizontal pairs) for 10x10 planar array recordings and 112 bipolar channels (56 vertical and 56 horizontal pairs) for 8x8 linear array recordings (in total 2628 channels over 9 sessions).

For all subsequent analyses, we epoched each trial data into a single 2s segment (from 0.5 to 2.5s after stimulus onset), which excludes initial transient responses. 2s is long enough for our analyses and is consistently available across all sessions. We excluded the datasets where we used 0.8s as a stimulus duration (and *F1* = 20Hz) which was mainly reported in [16]. All the experiments we report here come from experiments with *F1* = 23Hz and *F2* = 200Hz.

## Analysing power

By $\log P^{i,t,[g,h]}(f)$, we denote the logarithm (base 10) of power at frequency, *f*, of the LFP over 2s for channel *i* in trial *t*, where the amplitude of *F1* = 23Hz and *F2* = 200Hz was *g* and *h* [µm]. Hereafter, we simply call "power" to mean "logPower" for all subsequent analysis and statistics. We analyze the response frequency *f* from 0 to 250Hz. To obtain $\log P^{i,t,[g,h]}(f)$, we used *mtspectrumc.m* from Chronux Toolbox [18] with one taper. Together with the 2s time window length, this gives a half bandwidth of 0.5Hz (= (k+1)/2T; here, *k* is the number of tapers and *T* is the length of time window). The mean logPower $\overline{\log P}^{i,[g,h]}(f)$ is the mean across all trials in each stimulus condition. We use $\underline{X}$ to denote mean of *X* across trials. We had several channels

**Table 1. Summary of the experiments.**

| Session ID | Location of vibratory stimulation | 23Hz stimulus amplitudes [µm] | 200Hz stimulus amplitudes [µm] | # of trials per condition | Stimulus duration [s] |
|---|---|---|---|---|---|
| 1–1 | Contralateral D5 | 0, 10, 20, 40, 79, 159 | 0, 1, 2, 4, 7, 15 | 10 | 4 |
| 1–2 | Contralateral D4 | 0, 10, 20, 40, 79, 159 | 0, 1, 2, 4, 7, 15 | 10 | 4 |
| 1–3 | Contralateral D5 | 0, 19, 40, 79, 159 | 0, 2, 4, 7, 15 | 15 | 4 |
| 2–1 | Contralateral D4 | 0, 40, 79, 159 | 0, 4, 7, 16 | 15 | 3 |
| 2–2 | Contralateral D5 | 0, 40, 79, 159 | 0, 4, 7, 16 | 15 | 3 |
| 2–3 | Contralateral central pad | 0, 40, 79, 159 | 0, 4, 7, 16 | 15 | 3 |
| 2–4 | Contralateral central pad | 0, 19, 40, 79, 159 | 0, 4, 7, 16, 31 | 10 | 4.4 |
| 2–5 | Contralateral central pad | 0, 19, 40, 79, 159 | 0, 4, 7, 16, 31 | 10 | 4.4 |
| 2–6 | Contralateral D4 | 0, 19, 40, 79, 159 | 0, 4, 7, 16, 31 | 10 | 4.4 |

The details of the stimulus configuration per session. Each row has a distinct session ID. The first digit of the session ID refers to the cat ID. Within the same cat, we did not move S1 recording locations, but in some cases, we moved S2 recording locations. The second digit refers to different vibratory stimulation locations (e.g., pad, Dx finger, etc). We recorded from 2 cats over 9 sessions. 0 stimulus amplitude means no vibration at that frequency.

whose power happened to be 0 in some trials in some frequencies, which resulted in negative infinity after log transform. We removed these channels from further analysis.

We define signal to noise ratio (log$SNR$) of the response, log$SNR^{i,t,[g,h]}(f)$, by subtracting the logPower at the neighbouring frequencies of $f$ [Hz](denoted by a set $F'$ with $F' = \{f' \mid f\text{-}3 < f' < f\text{-}1$ or $f\text{+}1 < f' < f\text{+}3\}$) from the logPower at $f$;

$$log\ S\ N\ R^{i,t,[g,h]}(f) = log\ P^{i,t,[g,h]}(f) - \frac{1}{N_{F'}}\sum\nolimits_{f'\ \in f'} log\ P^{i,t,[g,h]}(f') \tag{1}$$

where $N_{F'}$ is the number of the neighbouring frequencies. log$SNR$ is a standard measure in SSEP literature [4] and facilitates comparisons of the signal quality across different recording modalities, brain regions, animals and paradigms. As our input stimulus contained sinusoidal tactile modulation at $F1$ = 23Hz and $F2$ = 200Hz, we expected that there would be log$SNR$ in the corresponding tagged frequencies $f1$ = 23Hz and $f2$ = 200Hz. Further, we expected that nonlinear processing to result in log$SNR$ in the harmonic frequencies (i.e., the integer multiple of the tagged frequencies, $n1^*f1$). As we restricted our analyses within 0 to 250Hz, we analyzed 9 harmonic frequencies: 46, 69, 92, 115, 138, 161, 184, 207, and 230Hz with $n1$ = 2, 3, 4, 5, 6, 7, 8, 9 and 10, respectively. We further expected that nonlinear integration of the two stimuli to result in log$SNR$ at the intermodulation frequencies, $n1^*f1+n2^*f2$, where $n1$ and $n2$ are both nonzero integers. When $n2$ = 1, this resulted in 10 intermodulation frequencies: 16, 39, 62, 85, 108, 131, 154, 177, 223, and 246Hz, with $n1$ = -8, -7, -6, -5, -4, -3, -2, -1, 1, and 2, respectively. In total, we considered 21 frequencies (2 tagged, 9 harmonic, and 10 intermodulation frequencies). Note that the closest neighboring frequencies among tagged, harmonic, and intermodulation frequencies are 7Hz (e.g., $f1$ = 23Hz vs (-8)$^*f1+f2$ = 16Hz), which is much bigger than the half bandwidth of 0.5Hz (see above).

Here, log$SNR$ assumes that frequencies outside of tagged, harmonic, and intermodulation frequencies reflect the noise and do not reflect the processing of the stimuli [4]. We were, at least initially, agnostic about such an assumption (See Discussion on this issue).

We also define vibration evoked logPower, $VELogP^{i,t,[g,h]}(f)$, as logPower minus mean logPower across trials in which the stimulus probe touched the cat's paw without any vibration (i.e. $[g, h]$ = $[0, 0]$). That is,

$$VELog\ P^{i,t,[g,h]}(f) = log\ P^{i,t,[g,h]}(f) - \frac{1}{NT}\sum\nolimits_{t'\ \in T} log\ P^{i,t',[0,0]}(f) \tag{2}$$

where $T$ is a set of $[g, h]$ = $[0, 0]$ trials and $N_T$ is the number of such trials. This captures how presenting vibration makes a difference in logPower from the no-vibration condition. Like log$SNR$, we expected that there would be $VELogP$ in the target, harmonic, and intermodulation frequencies.

In addition to the tagged, harmonic, and intermodulation frequencies, we observed broad high-gamma responses [19–22]. In our preliminary analyses, where we examined the responses of all frequencies that are outside of the tagged, harmonic, and intermodulation frequencies, we observed strong modulation of the high-gamma band (50-150Hz) power with some dependency on the amplitude of $F1$ and $F2$ vibration. If there is a strong evoked broadband response, the log$SNR$ measure would not detect such an increase. Broad and uniform increase of power will cancel out due to the neighborhood-subtraction procedure in logPower, resulting in log$SNR(f)$ = 0. To analyze the observed high-gamma responses, we used a different measure, high gamma power (HGP), in which we selectively averaged the power in the high gamma range excluding any contribution from tagged, harmonic and intermodulation frequencies. For this purpose, we first define a set of frequencies, $f'$, between 50 and 150Hz, which are outside of +/- 0.5Hz around the tagged, harmonic, and intermodulation frequencies. We

define high gamma power (log*HGP*) as the mean vibration evoked logPower (*VELogP*) across *f'* as follows:

$$log \, H \, G \, P^{i,t,[g,h]} = \frac{1}{N_{f'}} \sum\nolimits_{50 < f' < 150} VELog \, P^{i,t,[g,h]} \, (f') \tag{3}$$

where $N_{f'}$ is the number of the frequencies considered for HGP. Note that raw log values are multiplied by 10 and values introduced here are shown in [dB].

## Statistical analysis

To investigate the effect of the amplitude of vibration stimuli at *F1* and *F2*, we performed a two-way analysis of variance (ANOVA) using the MATLAB function, *anova2.m* (MATLAB R2019b). We modelled the dependent variable *y*, where *y* can be log*P*(*f*), log*SNR*(*f*), or *VELogP* (*f*), at channel *i*, and response frequency *f*:

$$y(i, f) = cF_1 \times aF_1 + cF_2 \times aF_2 + cF_1F_2 \times aF_1 \times aF_2 + \varepsilon \tag{4}$$

where *aF1* and *aF2* are the amplitude of *F1* = 23Hz and *F2* = 200Hz vibration stimuli (e.g., [*aF1*, *aF2*] = [0, 0], [0, 40]. . . . See Table 1) and *cF1*, *cF2*, *cF1F2* are the coefficient that minimizes the error term, $\varepsilon$, in the ANOVA framework. We performed the ANOVA to obtain *F*-statistics and *p*-value separately for each channel, *i*, and response frequency, *f*, for two main effects and their interaction, by testing the significant departure of *cF1*, *cF2* and *cF1F2* from zero.

We performed the above ANOVA for frequencies from 0Hz to 250Hz for each channel. To correct for multiple comparisons, we used false discovery rate [23] implemented as *fdr.m* in eeglab for MATLAB [24]. We set the false discovery rate as *q* = 0.05 to determine a corrected *p*-value threshold for significance by pooling across all frequencies, bipolar channels, sessions, somatosensory areas, and three effects (two main effects and their interaction).

## Results

### Steady state evoked potentials (SSEP) at a channel in the maximal stimulation condition

We used the steady state evoked potentials (SSEP) paradigm to probe the properties of population-level neuronal responses in cat's primary and secondary somatosensory areas (S1 and S2) under anesthesia. While we recorded local field potentials (LFPs) from planar array in S1 and linear electrode array in S2 (Fig 1A), we applied vibratory stimuli in low (*F1* = 23Hz) and high (*F2* = 200Hz) frequency around the cats' paw (Fig 1B and Table 1). In Fig 1C, we show a sample LFP response from one highly responding bipolar channel in S1 in Session 2–2. In the figure, we show the average of the bipolar data across 15 trials in which we applied the largest amplitude in the session, that is, stimulus condition [*F1*, *F2*] = [159,16]. Hereafter, we represent a combination of the vibration amplitude *X* and *Y* [μm] for frequency *F1* and *F2* as [*F1*, *F2*] = [*X*, *Y*].

To characterize neural responses at tagged, harmonic and intermodulation frequencies, we first computed the power spectra from 0 to 250Hz for each trial in each stimulus condition (see Methods) in total from 2628 channels. Fig 1D shows the power spectrum from the same channel in S1 as in Fig 1C (the average across 15 trials from the maximum stimulus condition, [*F1*, *F2*] = [159, 16], in Session 2–2). The frequencies of interest are shown: *f1* = 23Hz fundamental and harmonic frequencies with red vertical line, *f2* = 200Hz fundamental frequency with green vertical line and their intermodulations with blue vertical line. In the figure, we

observe several peaks, but $1/f$ distribution is dominant. To isolate the significance of responses at the frequencies of interest, we quantified the power increase from the baseline. First, following the SSEP literature [4], we used nontagged frequencies as the baseline. We refer to the measurement as signal to noise ratio (log*SNR*). Second, we used the no-stimulus condition, i.e. [*g*, *h*] = [0, 0], as the baseline. We refer to the measurement as vibration evoked logPower (*VELogP*; see Methods for the details). log*SNR* and *VELogP* were computed per trial, stimulus condition, and channel. Fig 1E and 1F show the example of log*SNR* and *VELogP* from the same channel as in Fig 1C and 1D (the average across 15 trials at [*F1*, *F2*] = [159, 16] in Session 2–2). These formats clarify the peaks of log*SNR* and *VELogP* at the frequencies of interest. As we will demonstrate, these observations were not unique to one particular channel, but generalized across many channels. Note that these results were not straightforwardly expected from our previous study with SUA and MUA in the same data set [16]. We will discuss the implication of each point in Discussion.

## Unexpected main effects and interactions of *F1* and *F2* across frequencies

Next, we quantified the response dependency on the magnitude of *F1*, *F2* and their interactions by performing two-way ANOVA on log*P*, log*SNR* and *VELogP*. The statistical analyses revealed that responses at the frequencies of interest were not observed across all channels and specific to some channels (Fig 2). For example, we found that some channels responded at *f1* (or *f2*), whose response magnitude depended on only the stimulus magnitude of *F1* (or *F2*) (Figs 3 and 4). However, we also found that some channels responded at *f1* (or *f2*), whose response magnitude depended on the stimulus magnitude of the "other" frequency, e.g., *F2* (or *F1*) (Fig 5). Further, we found that the assumption of log*SNR* computation, that is, the use of nontagged frequencies as the baseline was violated in some notable cases. This is hinted at by the difference between log*SNR* and *VELogP* around $50 < f < 150$Hz in Fig 2B (black line). As we elaborate in Fig 6, we found this effect only in the channels in S2 and at $50 < f < 150$Hz, so-called high gamma range.

## Three major response types observed

Using two-way ANOVA, we statistically evaluated the proportion of the 2628 channels that exhibited significant modulation in log*P*, log*SNR*, and *VELogP*. The factors were the input amplitude of *F1* = 23Hz and *F2* = 200Hz vibration. This analysis revealed that response at a frequency was mainly classified into three types: 1) response modulated by only *F1* amplitude (Fig 2A), 2) one modulated by only *F2* amplitude (Fig 2B), and 3) one modulated by the two amplitude and their interaction (Fig 2C). In addition, we found that the results from log*SNR* were different from those from log*P* and *VELogP*. In particular, while we observed modulation at $50$Hz$< f < 150$Hz in log*P* and *VELogP*, this modulation was not observed in log*SNR*. When it comes to *VELogP*, the results were exactly the same as ones of log*P* because of our definition of *VELogP*.

Fig 2A–2C respectively show the percentages of channels showing type 1), 2) and 3) significance for log*P*, log*SNR* and *VELogP* at each frequency. The frequencies of interest are shown with red (23Hz fundamental and harmonic frequencies), green (200Hz), and blue (intermodulation frequencies). The first type of response (Fig 2A) is found predominantly at the integer multiples of *f1* = 23Hz (~4%) and some intermodulation frequencies (~1%). Unlike the first type, the second type (Fig 2B) is mainly found at *f2* = 200Hz (~12%). Interestingly, this type is also found at $50 < f < 150$Hz (~2% on average), so-called high gamma range. Finally, the third type (Fig 2C) is predominantly found at *f1* = 23Hz (~6%) and 2*f2* = 46Hz (~7%), but also observed across some harmonic and intermodulation frequencies as well as *f2* = 200Hz (~2%).

In the next few sections, we will show exemplar log*SNR* responses of the three response types from Session 2–2. (See S4–S6 Figs for exemplar *VELogP* responses.)

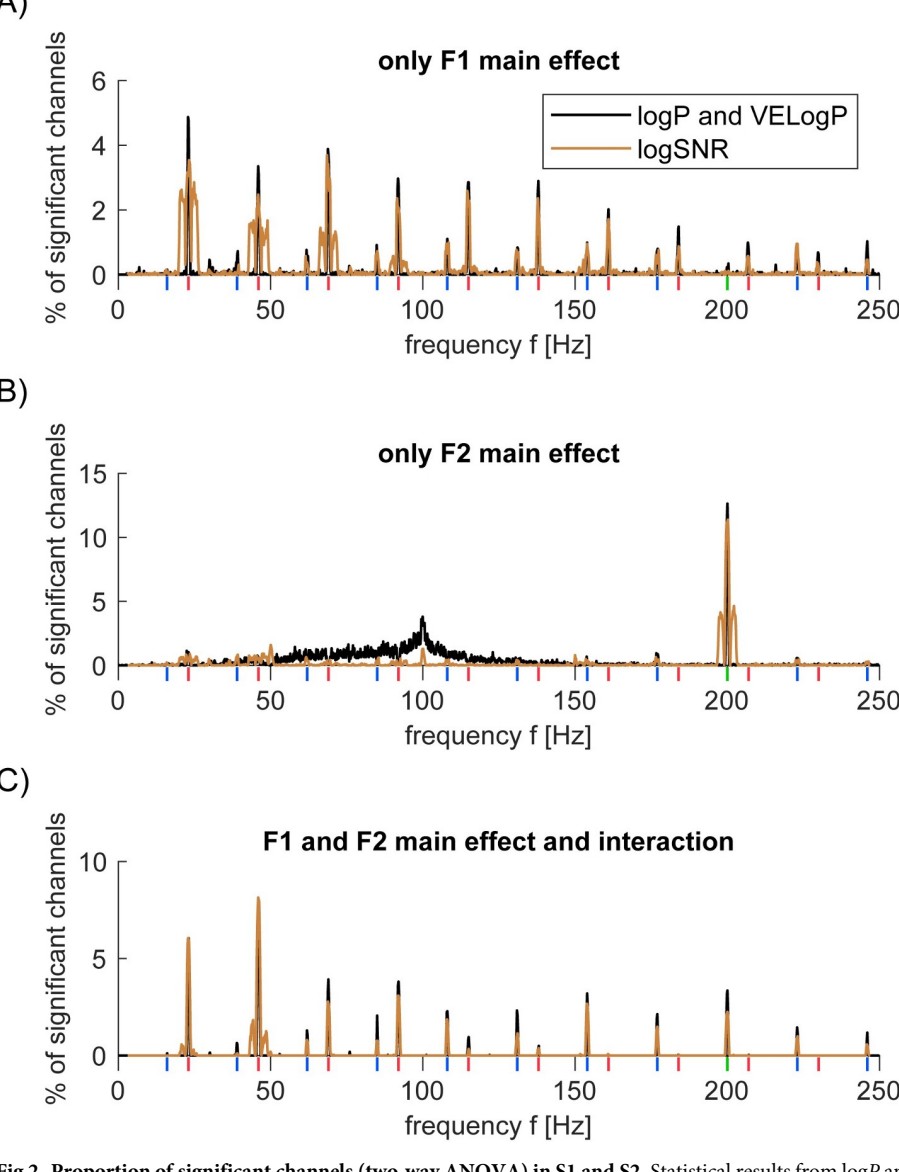

**Fig 2. Proportion of significant channels (two-way ANOVA) in S1 and S2.** Statistical results from log*P* and *VELogP* are identical and shown in black. Results from log*SNR* are shown in brown. Vertical lines beneath *x*-axis indicate *F1* fundamental and harmonic (red), *F2* fundamental (green), and intermodulation frequencies (blue). (A-C) % of channels that were deemed as significant according to two-way ANOVA only for the main effects of the amplitude of *F1* (but not *F2* main effect nor *F1-F2* interaction; A), *F2* = 200Hz main effect only (but not *F1* main effect nor *F1-F2* interaction; B) and both main effects of *F1* and *F2* as well as *F1-F2* interaction (C). We also tried to see a 'gain control' type effect by showing log*SNR* as functions of stimulus amplitudes (S1–S3 Figs). Unfortunately, the results were not clear possibly due to 1) variations in experimental conditions (Table 1) and 2) spatial specificity in the neural response (See Figs 3–5, below).

## Exemplar responses at *f1* (= 23Hz) only dependent on the magnitude of *F1* (= 23Hz)

In Fig 3, we show exemplar responses at *f1* = 23Hz which are modulated by only the magnitude of stimulus vibration at *F1* = 23Hz. Fig 3A depicts log*SNR* around 23Hz at bipolar channel 131 in S1 in Session 2–2. The channel demonstrates a significant main effect of *F1*

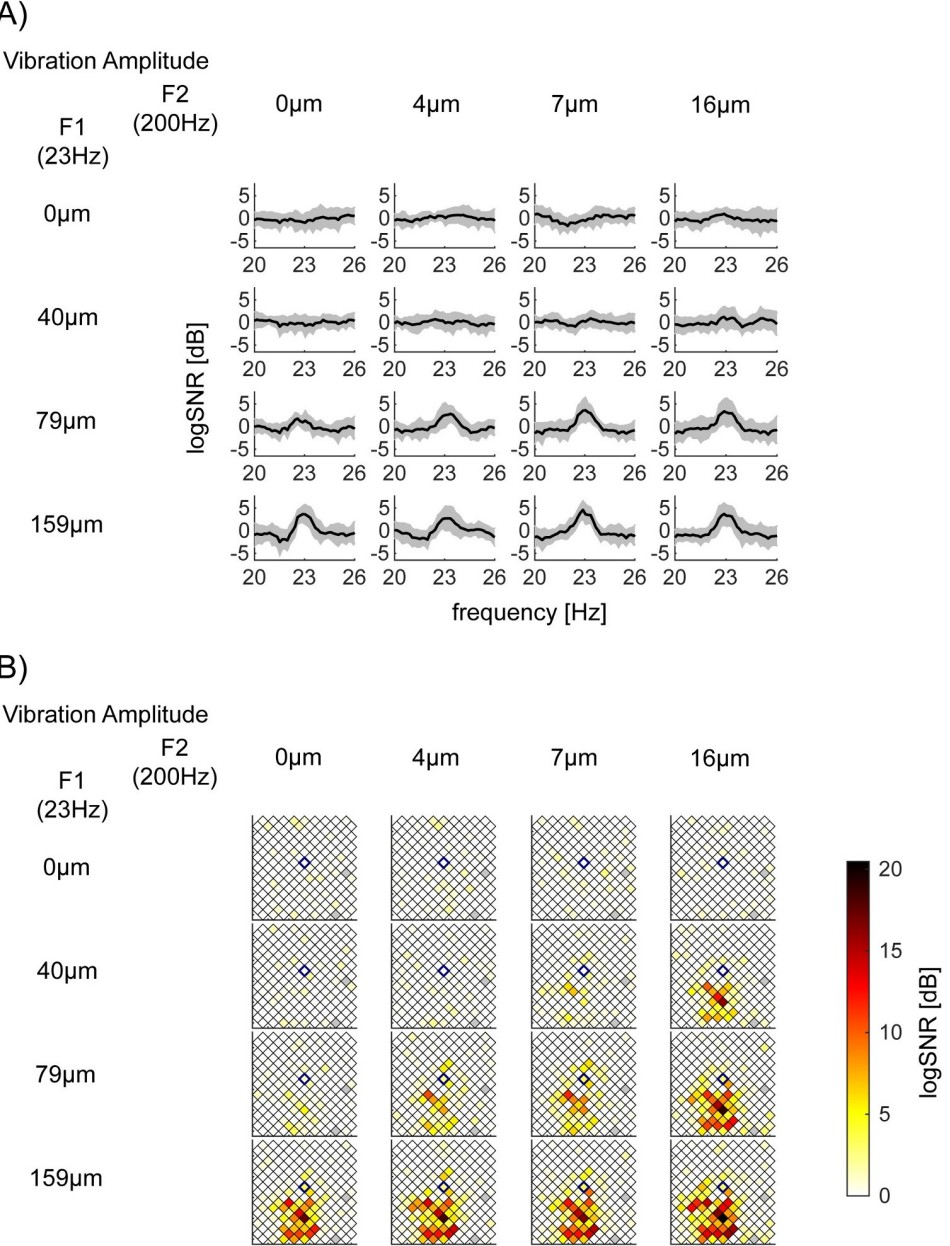

**Fig 3. Exemplar log*SNR* at *f1* = 23Hz depend on the vibration amplitude of *F1* = 23Hz.** 16 panels are arranged so that the row and column encodes the input amplitude of *F1* = 23Hz (from 0 to 159μm) and *F2* = 200Hz (from 0 to 16μm), respectively. (A) log*SNR* of S1 bipolar channel 131, whose location in the Utah array in (B) is identified with a blue diamond, (Session 2–2). This channel's responses at *f1* = 23Hz showed a significant main effect of *F1* = 23Hz amplitude, but neither the main effect of *F2* = 200Hz nor their interaction. *p*-value (*F1*, *F2*, interaction) = ($<10^{-5}$, 0.054, 0.52) with the corrected threshold 0.00019. *y*-axis of each subplot is the mean log*SNR* with standard deviation across 15 trials. *x*-axis is the response frequency *f*, around *f* = 23Hz. Note that, as we considered a set of frequencies *F'* = {*f* | *f*-3<*f*'<*f*-1 or *f*+1<*f*'<*f*+3} as the neighboring frequencies for the log*SNR* computation, log*SNR* is smoothed and has a lower spectral resolution than the half bandwidth of 0.5Hz. (B) Spatial mapping of log*SNR* at *f1* = 23Hz across all channels in S1 (Session 2–2). Each square represents one of the bipolar re-referenced channels. The center of the square is plotted at the middle point between the original unipolar recordings of the 10x10 array. Squares with gray indicate channels, which were removed from the analysis (see Methods).

= 23Hz amplitude ($p < 10^{-5}$), but neither the main effect $F2$ = 200Hz ($p$ = 0.054) nor their interaction ($p$ = 0.52; the corrected threshold 0.00019). Fig 3A shows that log$SNR$ at 23Hz was significantly modulated as a function of $F1$ = 23Hz amplitude from 0 to 159μm (from the top to the bottom row) but not across the amplitude of $F2$ = 200Hz amplitude from 0 to 16μm (from the left to the right column). Fig 3B shows the spatial distribution of log$SNR$ for all channels in S1 during Session 2–2. We use a blue diamond to mark the channel shown in Fig 3A.

## Exemplar responses at *f2* (= 200Hz) only dependent on the magnitude of *F2* (= 200Hz)

Fig 4 shows exemplar responses at *f2* = 200Hz which are modulated by only the magnitude of stimulus vibration at *F2* = 200Hz. Fig 4A depicts log$SNR$ around 200Hz for bipolar channel 36 in S1 in Session 2–2. The channel demonstrates a significant main effect of $F2$ = 200Hz amplitude ($p < 10^{-5}$), but not the interaction ($p$ = 0.14). The main effect of $F1$ = 23Hz ($p$ = 0.00043) does not survive the correction for multiple comparisons (0.00019). Fig 4B shows the spatial distribution of log$SNR$ for all channels in S1 during Session 2–2. It is clear that while many channels increase their log$SNR$ as the amplitude of 200Hz increases (from the left to the right column), a cluster of channels above bipolar channel 36 (blue diamond) also increases log$SNR$ at 200Hz as the amplitude of 23Hz increases (from the top to the bottom row).

## Exemplar responses at *f1* dependent on the magnitude of *F1* and *F2* and their interaction

Fig 5 shows exemplar responses at *f1* = 23Hz which are modulated by magnitude of stimulus vibration at both *F1* and *F2* and their interaction. Fig 5A depicts log$SNR$ around *f* = 23Hz for bipolar channel 158 in S1 in Session 2–2. The channel demonstrates the significant main effects of both 23Hz ($p < 10^{-5}$), 200Hz ($p < 10^{-5}$) as well as interaction ($p < 10^{-5}$) with the corrected threshold 0.00019. Fig 5B shows the spatial distribution of log$SNR$ for all channels in S1.

## Violation of SNR computation in the high gamma band in S2, but not in S1

The steady state evoked potentials (SSEP) paradigm assumes that the responses at the "non-tagged" frequencies (i.e., outside of the tagged, harmonic and intermodulation frequencies) do not represent stimulus processing, thus, it can be considered as the noise or baseline. The log$SNR$ measure we presented so far rests on this assumption. To contrast the difference between log$SNR$ and *VELogP* responses, we present the data in Figs 3–5 in the format of *VELogP* in S4–S6 Figs.

Fig 6A–6C show the *F*-statistics for *F1* main effect, *F2* main effect, and their interaction respectively from our two-way ANOVA on *VELogP* (not log$SNR$) across all frequencies from 0 to 250Hz. Each of them shows the 95th, 90th, and 50th percentile for *F*-statistics for $N$ = 1620 channels in S1 and $N$ = 1008 channels for S2. For these figures, we removed +/-0.5Hz around the fundamental, harmonic and intermodulation frequencies in order to focus on the modulation of responses at the nontagged frequencies. Blue and pale blue lines for S1 in Fig 6A–6C demonstrate that S1 does not show any effects outside the fundamental, harmonic, and intermodulation frequencies. Red and pale red lines for S2 are very similar in Fig 6A and 6C. However, they are quite distinct in Fig 6B, which demonstrates a strong main effect of *F2* = 200Hz across HGB (from 50 to 150Hz) in some channels in S2. Fig 7A shows one exemplar response in bipolar channel 102 from Session 2–2 in S2, which shows clear main effects of *F2* = 200Hz

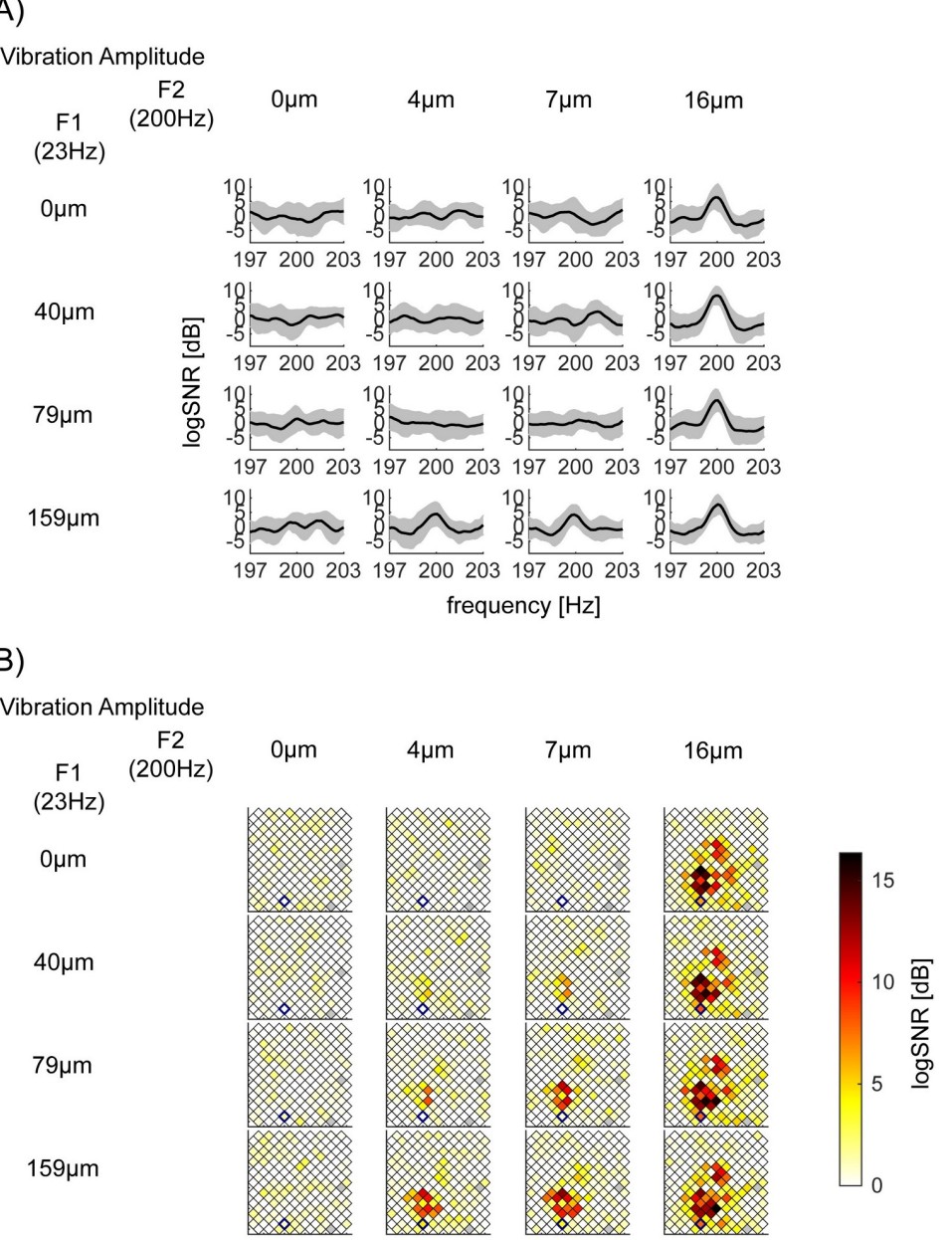

**Fig 4. Exemplar log*SNR* at *f2* = 200Hz depends on the vibration amplitude of *F2* = 200Hz.** This figure is shown with the same format as Fig 3. (A) log*SNR* of bipolar channel 36 in S1 (Session 2–2). This channel's responses at *f2* = 200Hz showed a significant main effect of *F2* = 200Hz amplitude, but neither the main effect of *F1* = 23Hz nor their interaction. *x*-axis is the response frequency *f*, around *f* = 200Hz. *p*-value (*F1*, *F2*, interaction) = (0.00043, $<10^{-5}$, 0.14) with the corrected threshold 0.00019. (B) Spatial mapping of log*SNR* at *f2* = 200Hz across all channels in S1 (Session 2–2).

across 50-150Hz. Fig 7B shows the spatial mapping of HGP in S2, which demonstrates the wide spread high gamma responses in S2. (See Methods for HGP.)

The result is quite unlike all the analyses we presented so far, where we see no clear differences between the two regions, S1 and S2. We will come back to the source of the differences and its implication in SSEP paradigm in Discussion.

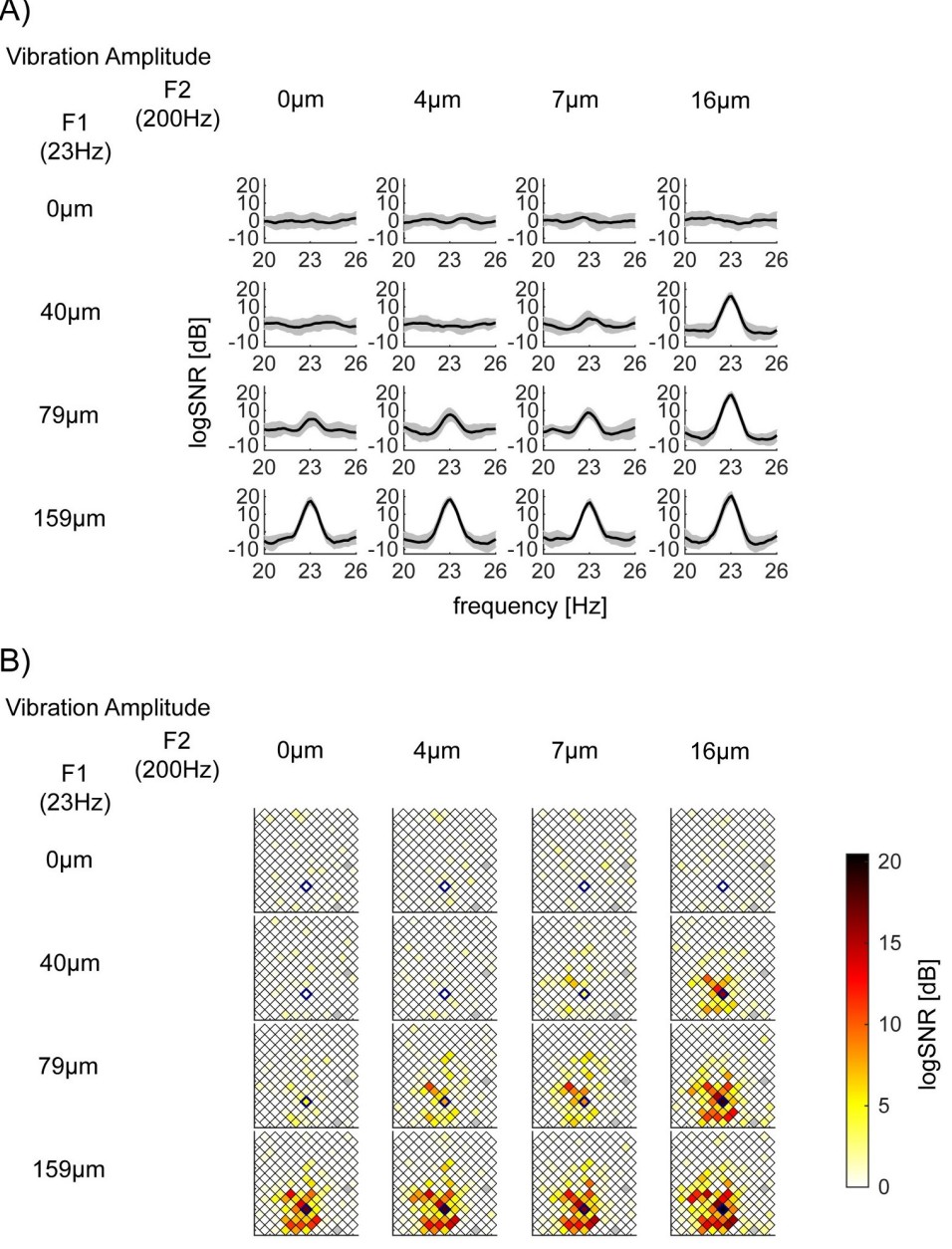

**Fig 5. Exemplar log*SNR* at *f1* = 23Hz depend on the vibration amplitude of *F1* = 23Hz, *F2* = 200Hz, and their interaction.** The same format as Fig 3. (A) log*SNR* of bipolar channel 158 in S1 (Session 2–2). This channel's responses at *f1* = 23Hz showed a significant main effect of *F1* = 23Hz amplitude, the main effect of *F2* = 200Hz, and their interaction. *x*-axis is the response frequency *f*, around *f* = 23Hz. *p*-values (*F1*, *F2*, interaction) are all $p < 10^{-5}$ with the corrected threshold 0.00019. (B) Spatial mapping of log*SNR* at *f2* = 200Hz across all channels in S1 (Session 2–2).

## HGP is sustained after the onset transient

HGP is typically observed as a transient response [22, 25], thus, the fact that we saw the HGP may be dependent on our choice of the duration of the response for computing the spectral power. To examine this, we analyzed the time course of the HGP. Here, we shortened the time window from 2 to 0.5s, thus, the half bandwidth increased from 0.5 to 2Hz. To avoid the

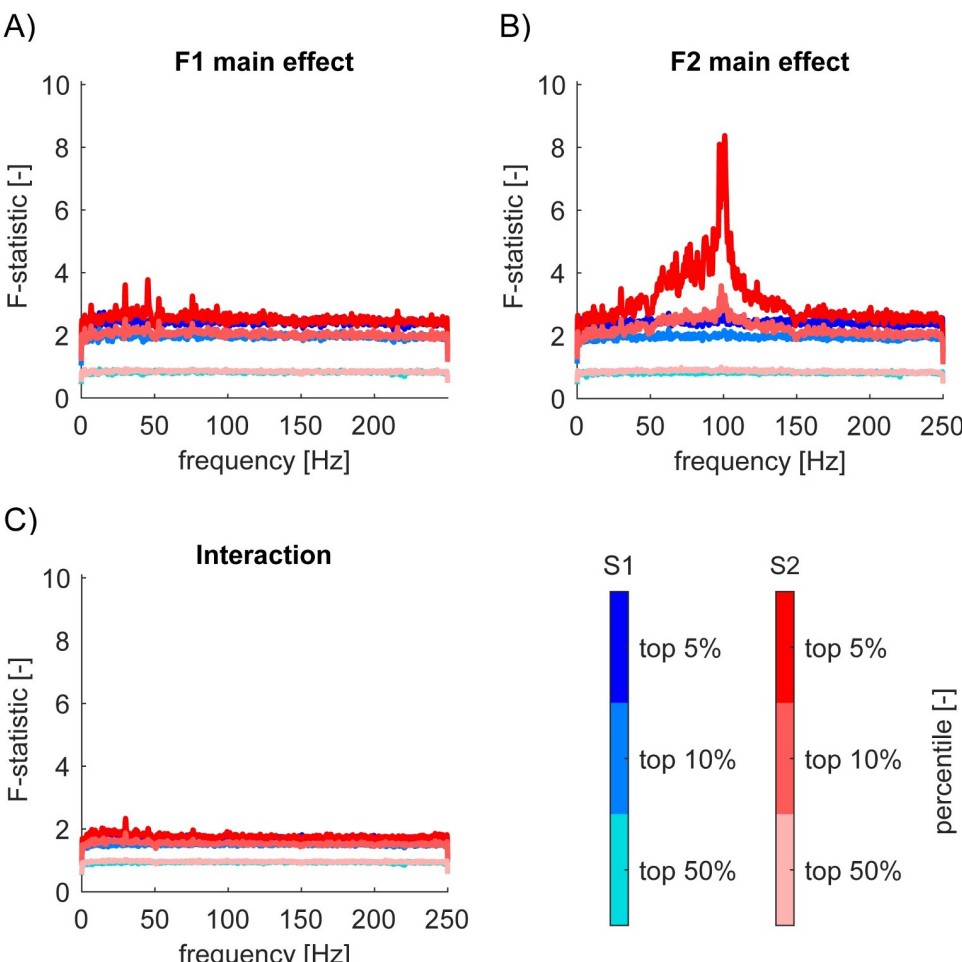

**Fig 6. Nontagged frequencies between 50-150Hz in S2 are modulated by *F2* = 200Hz vibratory amplitude.** (A)-(C) *F1* main effect (A), *F2* main effect (B), interaction (C) *F*-statistics from ANOVA performed on log*P* in S1 (shades of pale blue to blue) and S2 (shades of pale red to red). We plotted 95th percentile (top 5%), 90th (top 10%), and 50th (median). Some channels in S2 showed significant main effect of 200Hz vibratory amplitude in nontagged frequencies (+/-0.5Hz outside of the tagged, harmonic, and intermodulation frequencies, and outside of 50Hz, 100Hz, and 150Hz) from 50-150Hz. No main effects of 23Hz amplitude or interaction. We smoothed lines for a display purpose (1 data point per 1Hz). Note that [-] represents unitless.

influence of the tagged frequencies, we averaged the power from 50 to 150Hz except for 2Hz around the tagged frequencies.

Based on the ANOVAs results, we searched for channels where HGP was modulated by the amplitude of *F2* stimulus vibration. Specifically, we used *F*-stats results from ANOVAs performed on log*P*. We took the mean of the *F*-stats across frequencies from 50 to 150Hz, avoiding the frequencies of interest +-2Hz for each channel and selected the top 10% of channels that showed larger mean *F*-stats.

Fig 8 shows the mean time course of HGP (black line) and responses at frequencies of interest around HGB from the top 10% channels. When computing the mean time course, we took only the maximum stimulus condition from each session. The HGP increase was prominent at the stimulus onset of the transient. After the transient, however, the HGP persisted and its magnitude was sometimes as strong as responses at harmonic and intermodulation frequencies. This sustained increase of HGP in nontagged frequencies violates the assumption for the

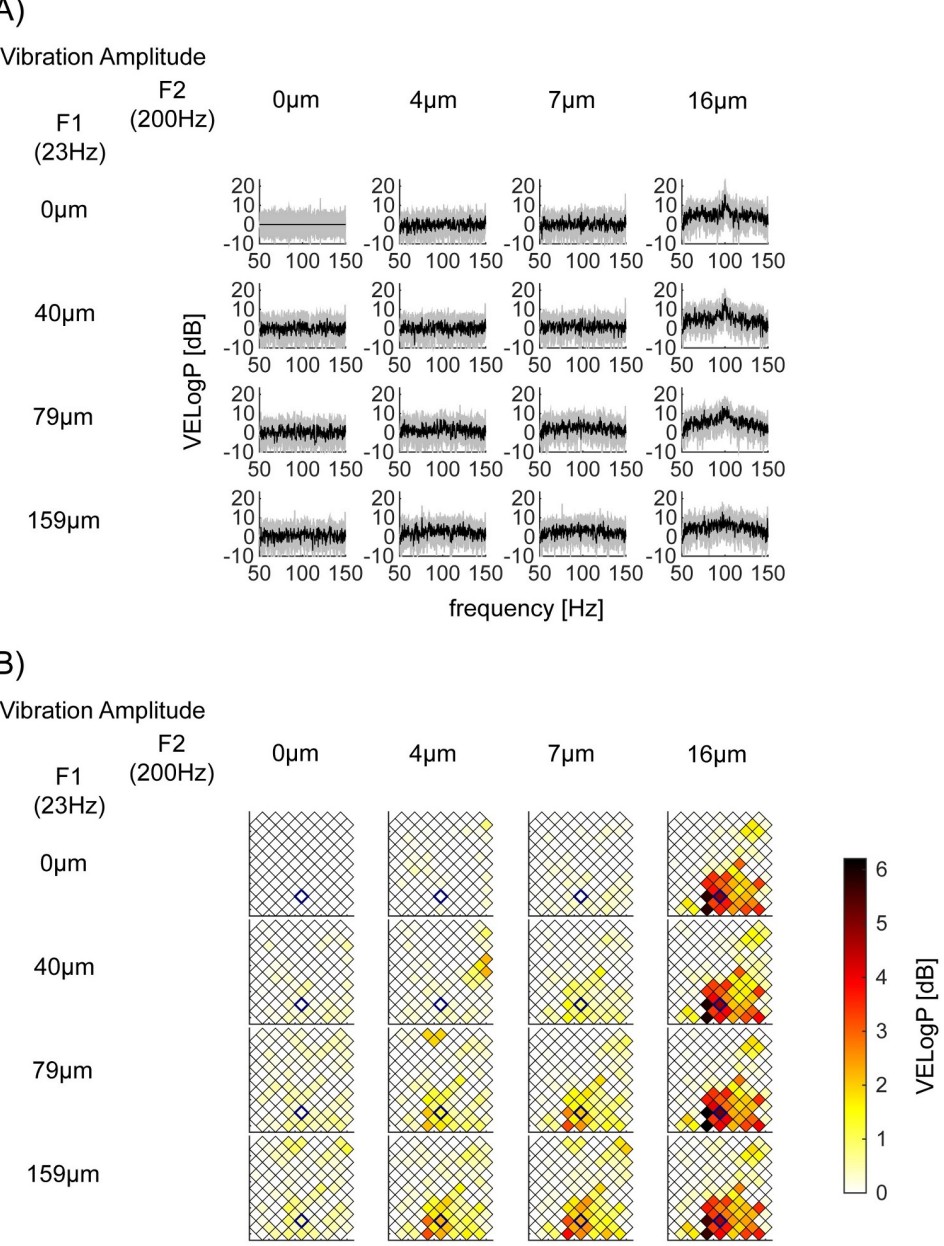

**Fig 7. Exemplar nontagged frequencies responses between 50-150Hz modulated by *F2* = 200Hz vibratory amplitude.** (A) Exemplar nontagged responses (*VELogP*) between 50-150Hz in bipolar channel 102 in S2 (Session 2–2). Note that the top left panel, which corresponds to the no- vibration condition, gives a flat line because *VELogP* is defined as the deviation of logPower from mean logPower of no vibration condition. It still has some variance across trials, shown as standard deviation (grey shading). (B) Spatial mapping of high gamma power (HGP) in S2 (Session 2–2). The location of bipolar channel 102 is located in the 8x8 array by a blue diamond. Color encodes the mean nontagged HGP in dB (our HGP measure, see Methods).

computation of signal-to-ratios; that is, nontagged frequencies do not reflect the stimulus processing and can serve as the baseline for the computation of log*SNR*. We will return to this issue in Discussion.

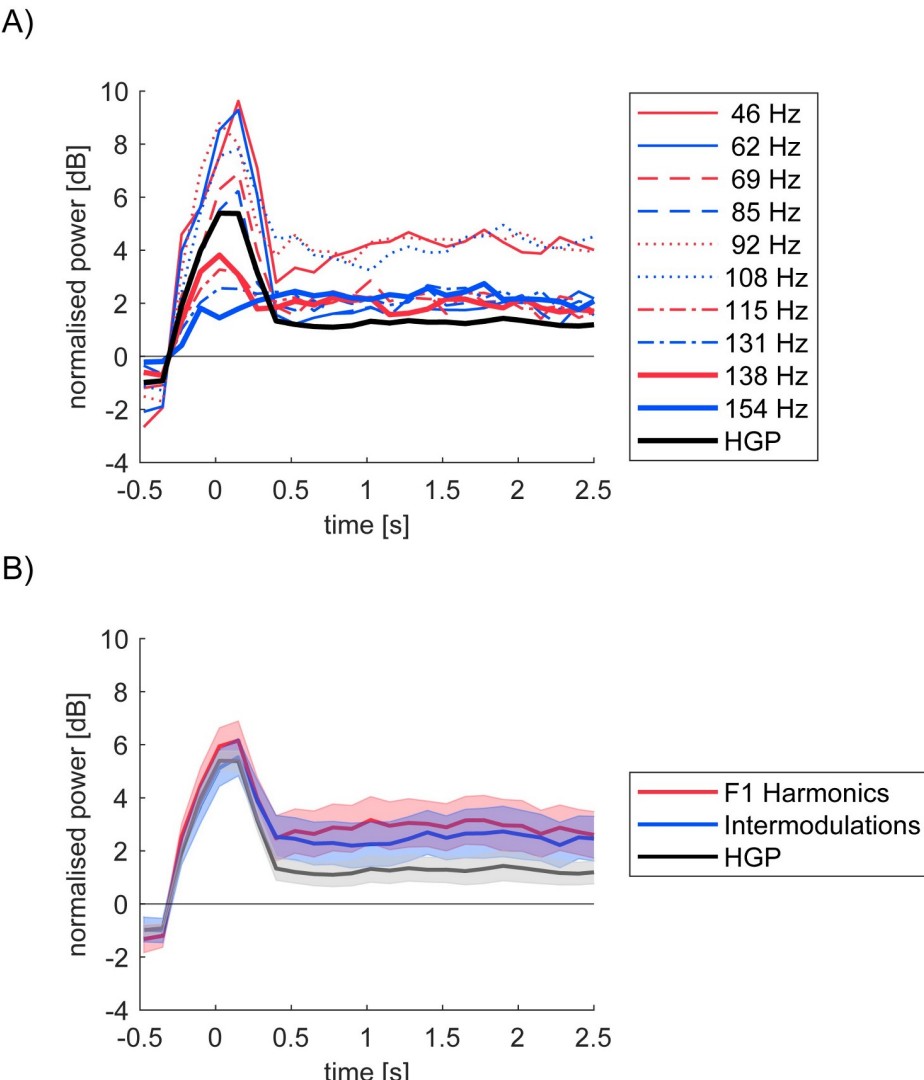

**Fig 8. Time course of the mean nontagged HGP and the frequencies of interest in S2.** (A) Evoked bandlimited power (half band width = 2Hz) around the stimulus onset. Mean log power is first averaged within the frequencies of interest for HGP (thick black), $f1$ harmonic (red), and intermodulation (blue) frequencies around HGB for each trial. Different line types for different frequencies (see the legend). Mean across trials per channel is further averaged across channels. The mean power during -0.5 to -0.25s for each frequency per trial is subtracted. (B) Summary of HGP, harmonic and intermodulation frequencies. Time courses for the harmonic (red) intermodulation (blue) in (A) are averaged for each category within each channel. Shading represents standard error of the mean across the top 10% channels selected as in Fig 6.

## Computational models to account for the observed nonlinear processing

As have been demonstrated so far, in addition to the targeted stimulus frequencies, $f1 = 23$Hz and $f2 = 200$Hz, we observed strong response modulation at the harmonic and intermodulation frequencies as a function of the vibration amplitude of $F1$ and $F2$ stimuli (Figs 1 and 2). Harmonic and intermodulation responses, especially the latter, have been taken as evidence of nonlinear processing in brains [15]. How such nonlinear processing is achieved by the neural circuitry in the brain, however, remains unclear.

In order to infer potential neural mechanisms which could give rise to the observed nonlinear responses, we used a computational modelling approach. Given the demonstrated biological plausibility [15, 26–28], here we focused on two types of nonlinear processing, that is, rectification and half squaring. For a sinusoidal input $X = \sin(2\pi^* f^* t)$, where $t$ is time and $f$ is input frequency, we modelled rectification nonlinearity as,

$$Rect(X) = \begin{cases} X & (\text{if } X > 0) \\ 0 & (\text{otherwise}) \end{cases} \tag{5}$$

and half squaring nonlinearity as,

$$HSq(X) = \begin{cases} X^2 & (\text{if } X > 0) \\ 0 & (\text{otherwise}) \end{cases} \tag{6}$$

Using an information theoretic model comparison approach, we selected the most plausible model among candidates, separately for those observed within S1 and S2.

As shown in Fig 9, these nonlinear operations alone do not generate responses at intermodulation frequencies while they do so when combined with further operations, such as multiplication. To reduce possible model architectures, we first visually inspected the data and the outputs from canonical models (Fig 9). We considered two classes of models; one based on combinations of *Rect* and the other based on *HSq*. For the *Rect* class, we considered *Rect(X)* +*Rect(Y)*+*Rect(XY)*+*Rect(X)Rect(Y)* as a full model and compared its reduced model by model comparison procedure using Akaike's Information Criteria (*AIC*) [29] (See Table 2 for all models). Fig 9 displays waveforms in the time-domain and spectra in the frequency-domain for each model class with coefficients set to be 1 for all terms.

In the fitting step, we used a simple line search optimization technique [30] to optimize coefficients for each term (for the details, see S1 Appendix). This optimization minimized the difference between the log*SNR* of the model output and the observed mean log*SNR* for a given channel across the frequencies of interest (Fig 10B). We adopted the mean log*SNR* across trials from the maximum vibration condition. For a given channel, we thus have 8 sets (4 for rectification and 4 for half squaring class) of the best fit coefficients and the minimum difference between model and actual log*SNR*.

To focus on a possible neuronal mechanism of harmonic and intermodulation nonlinear responses, we focused on the real channels that showed these nonlinear responses strongly. Specifically, we chose the top 10% channels for S1 (156 out of 1620 channels) and S2 (101 out of 1008 channels) in terms of the sum of log*SNR* across harmonic and intermodulation frequencies at the maximum stimulus condition.

Fig 10 illustrates exemplar model responses. Fig 10A shows log*SNR* for the real data (bipolar channel 43 in S1 in Session 2–2) and the optimized models while Fig 10B shows the difference between the real data and the optimized model at the frequencies of interest, that is, *f1*, *f2* fundamentals, harmonic, and intermodulation frequencies. For this channel, we see that log*SNR* at intermodulation frequencies at 177Hz and 223Hz are completely missed by the simplest model (i.e., *Rect(X)+Rect(Y)*), while they are explained by the rest of three non-linear models. However, *Rect(XY)* nonlinearity cannot account for log*SNR* at 154Hz and 246Hz. *Rect(X)Rect (Y)* is required to explain these components. We observed only slight improvement by the full model (i.e. *Rect(X)Rect(Y)* and *Rect(XY)*). *HSq* nonlinearity showed a different pattern of fitting from *Rect* nonlinearity (See S7 Fig). Specifically, in the case of *HSq* nonlinearity, adding a type of multiplication (i.e. *HSq(XY)*) did not change fitting performance at all.

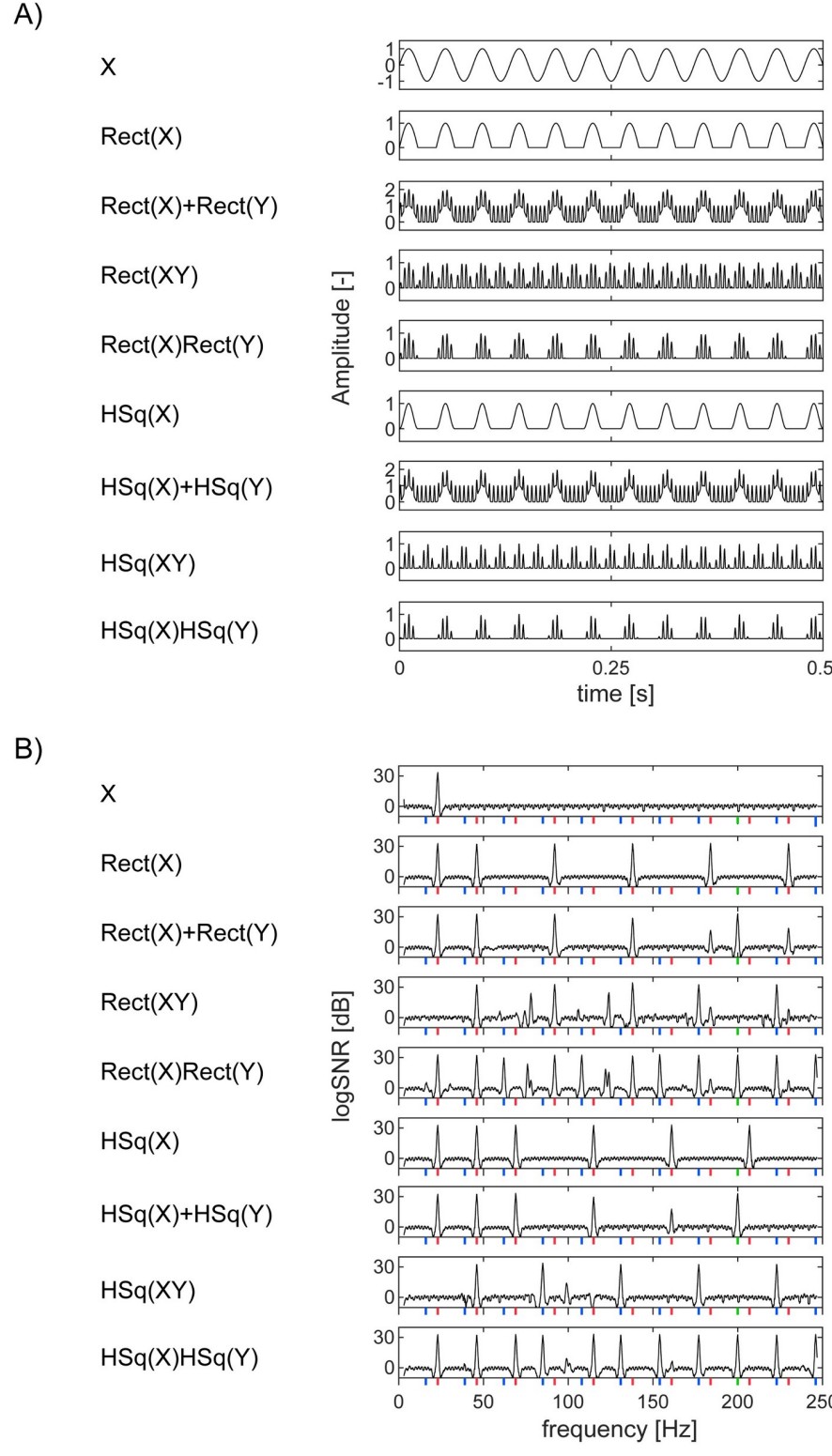

**Fig 9. Waveforms of modeled processes.** (A) Waveforms in the time domain (0 to 0.5s). *X* and *Y* are sinusoidal inputs at 23 and 200Hz, respectively. For the definitions of *Rect* and *HSq*, see the main text. (B) Spectra of each waveform in the frequency domain.

**Table 2. Models considered in analysis.**

| Model ID | Model |
|----------|-------|
| Rect-1 | $a^*Rect(X)+b^*Rect(Y)$ |
| Rect-2 | $a^*Rect(X)+b^*Rect(Y)+c^*Rect(XY)$ |
| Rect-3 | $a^*Rect(X)+b^*Rect(Y)+c^*Rect(X)Rect(Y)$ |
| Rect-4 | $a^*Rect(X)+b^*Rect(Y)+c^*Rect(XY)+d^*Rect(X)Rect(Y)$ |
| HSq-1 | $a^*HSq(X)+b^*HSq(Y)$ |
| HSq-2 | $a^*HSq(X)+b^*HSq(Y)+c^*HSq(XY)$ |
| HSq-3 | $a^*HSq(X)+b^*HSq(Y)+c^*HSq(XY)$ |
| HSq-4 | $a^*HSq(X)+b^*HSq(Y)+c^*HSq(XY)+d^*HSq(X)HSq(Y)$ |

This table shows all models we considered for our analysis. *a,b,c* and *d* are coefficients to be optimized.

Next, we summarized the population level statistics of the rectification model fitting. Fig 11 shows the cumulative probability distribution curves of the minimum difference across bipolar channels for each rectification model. The better a model is able to fit responses overall across channels, the quicker its cumulative distribution curve starts in the x-axis and reaches the top i.e. probability = 1 in the y-axis. (See S9 Fig for the half squaring class.)

To address the question of whether integrative computation is observed more frequently in S2 than S1, we compared the model fitting results between S1 and S2 (the left and right panel in Fig 11A). For S1 (the left panel), all curves appeared to similarly start in the x-axis and reach the top i.e. all models appeared to be almost equally fitted to responses in S1. For S2 (the right panel), on the other hand, some curves reached the top quicker than the others i.e. some models were able to fit better than the others. Specifically, the simplest *Rect(X)+Rect(Y)* model, depicted by the blue lines, has a more similar curve to the other models for S1 responses than for S2 responses. Adding another integrative nonlinear *Rect(XY)* component (orange lines) nearly saturated our model performance for S1, while it still left room for improvement for S2. Additional improvement was achieved when we considered the *Rect(X)Rect(Y)* component (purple and yellow), which led to an improvement especially in S2. We had a similar observation from the half squaring class except the fact that including *HSq(XY)* did not change fitting performance at all. (See S9 Fig)

To take into account the number of parameters used for each model class, we computed Akaike's Information Criterion (*AIC*) [29] for each model class within S1 and S2 (See S2 Appendix for the details). Fig 12 shows *AIC* for 8 models, demonstrating the fitting performance is best for the full rectification model for both S1 and S2 when taking into account the number of model parameters. Since adding *HSq(XY)* did not change the fitting performance, the models without *HSq(XY)* were better than the models with *HSq(XY)* when we corrected the bias.

Finally, we confirmed the statistical difference of the cumulative probability distributions between S1 and S2 for each model (Fig 11B). Kolmogorov-Smirnov tests confirmed a significantly better model fit ($p<10^{-5}$) for S1 than S2 for the simplest model (Fig 11B top left). Although the three components models resulted in better performance for S1 than S2, the difference was not as large as the simplest model ($p = 0.0051$ for the model with *Rect(XY)* in Fig 11B top right and $p = 0.024$ for the model with *Rect(X)Rect(Y)* in Fig 11B bottom left). For the full model, the difference in model fitting was not significant ($p = 0.072$ in Fig 11B bottom right). (All *p* values were Bonferroni corrected). Although adding an integrative nonlinear components *HSq(X)HSq(Y)* still showed a statistically significant difference between S1 and S2 by Kolmogorov-Smirnov (the right panel in S9 Fig), it attenuated the difference between S1 and S2 compared to the simplest model *HSq(X)+HSq(Y)* (the left panel in S9 Fig). These

A)

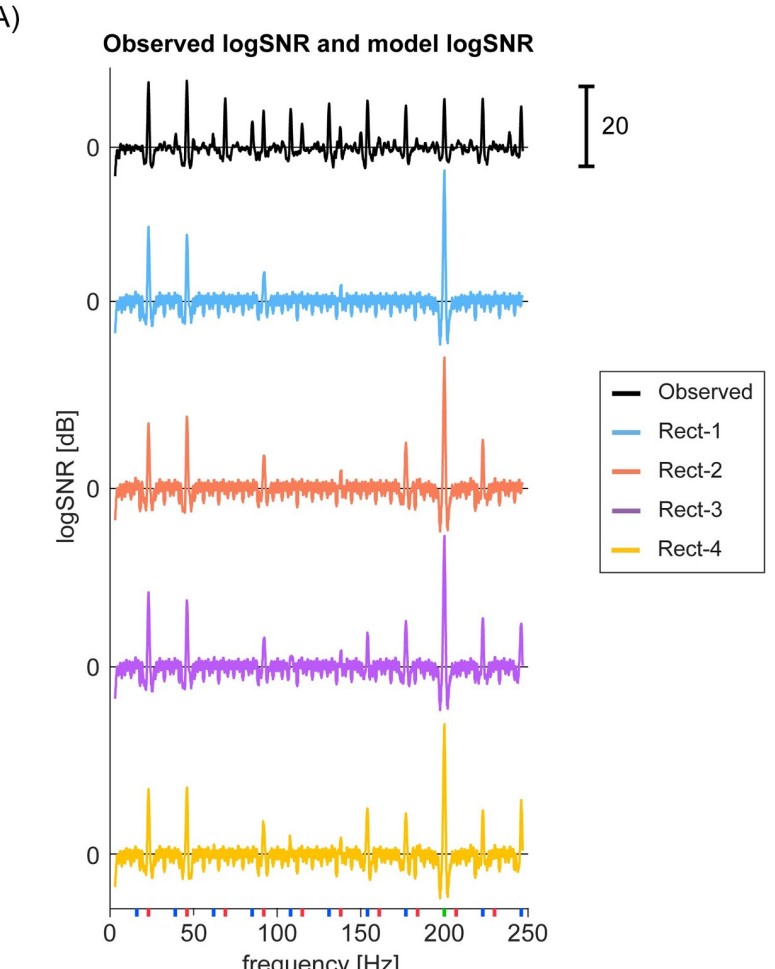

B)

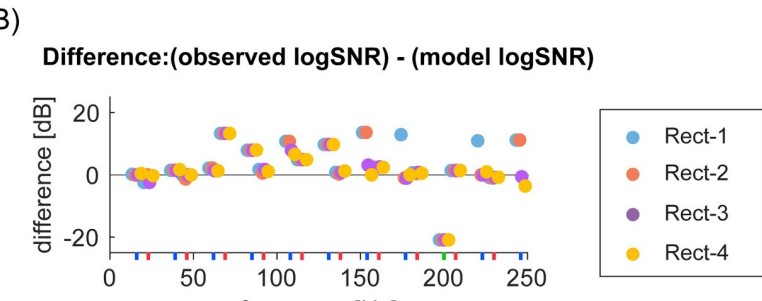

**Fig 10. Exemplar channel's observed and modelled responses.** (A) Bipolar channel 43 in S1 (Session 2–2)'s log*SNR* (black) is compared to the optimally fitted models from four model architectures based on *Rect* functions. The best parameters for each model are the following. Rect-1 (red): -0.023*Rect(X)*+1.0*Rect(Y)*, Rect-2 (blue): -0.98*Rect(X)* +58*Rect(Y)*-0.90*Rect(XY)*, Rect-3 (purple): -0.98*Rect(X)*+30*Rect(Y)*+0.094*Rect(X)Rect(Y)* and Rect-4 (yellow): 1.8*Rect (X)*+55*Rect(Y)*+0.48*Rect(XY)*-2.4*Rect(X)Rect(Y)*. (See S8 Fig for parameters from other channels.) The respective sums of log*SNR* differences from the observed log*SNR* across the frequencies of interest are 130, 105, 83 and 79. (B) Difference between the observed and the best model at the frequencies of interest. Color scheme is the same as in (A).

results suggest that the integrative nonlinear components played a significant role in explaining the variance of S2 channels to make the quality of our model fitting equivalent between S1

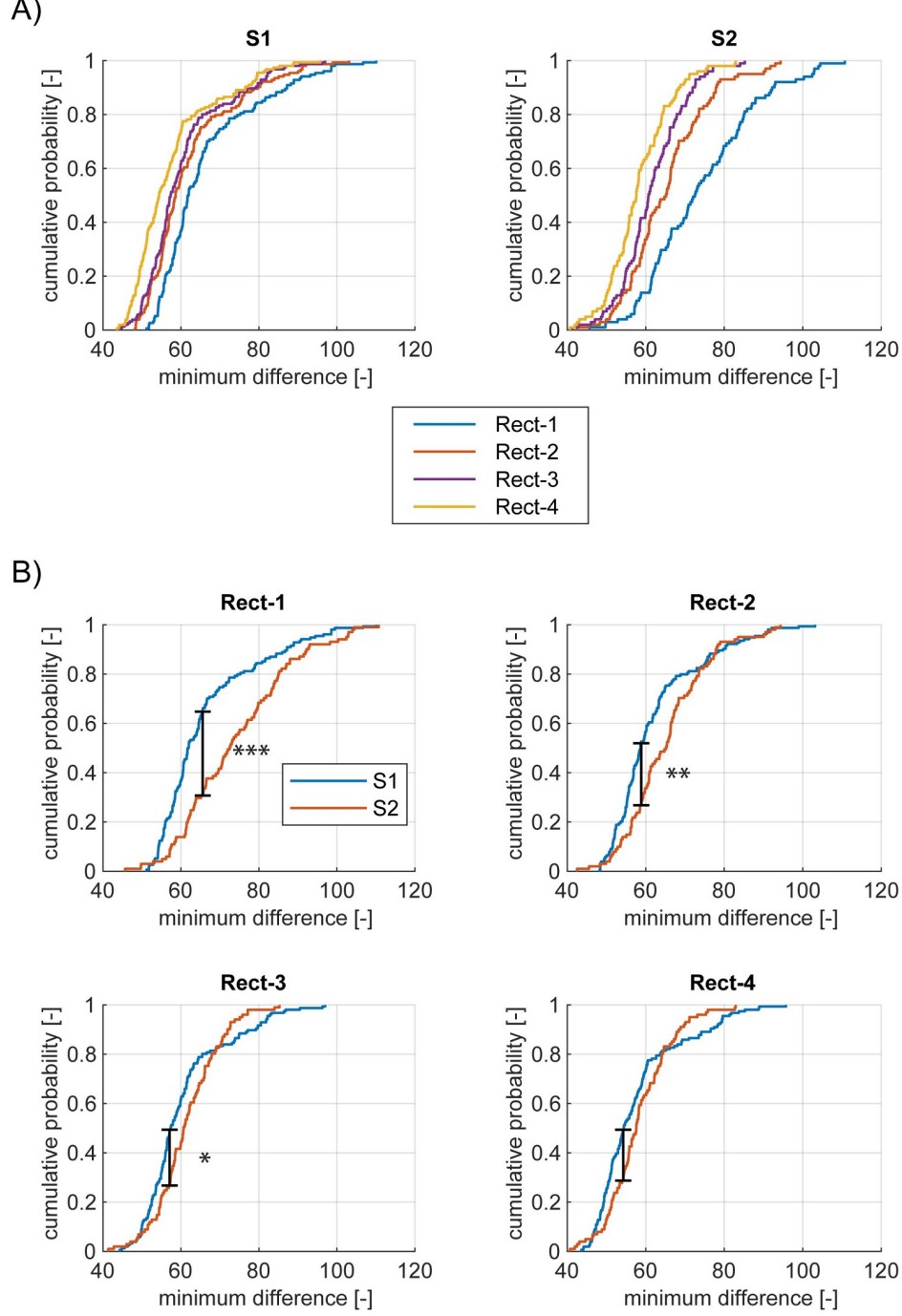

**Fig 11. Comparison of fitting performances across rectification models.** (A) Comparison of performance across four rectification models based on the cumulative probability distributions of the minimum difference for S1 and S2 separately. For a display purpose, we removed the worst 1% channels in S1 for two rectification models *Rect(X)+Rec(Y)* and *Rect(X)+Rect(Y)+Rect(XY)*. (B) Comparison of each model's performance between S1 and S2 based on the cumulative probability distributions of the minimum difference. *, **, and *** indicates $p<0.05$, $p<0.01$, and $p<0.001$ according to Kolmogorov-Smirnov tests.

and S2. These analyses confirm the presence of more complex integrative nonlinear interactions in S2, which is lacking in S1 as we elaborate in Discussion.

A)

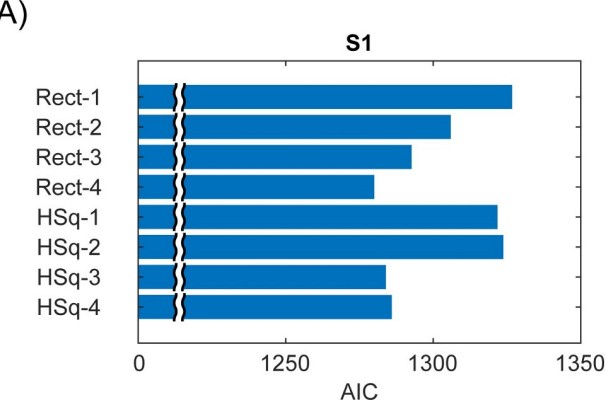

B)

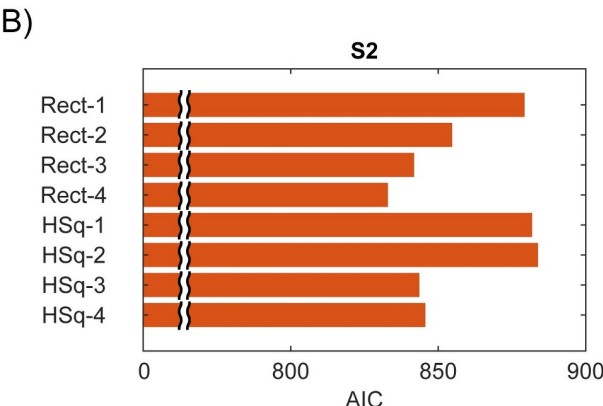

**Fig 12. Akaike's Information Criterion (*AIC*) for 8 models for S1 and S2.** The best model is the full rectification model for both S1 and S2.

## Discussion

In this paper, we examined local field potentials (LFPs) recorded from primary and secondary somatosensory cortex (S1 and S2) of anesthetized cats. Our four main findings are: 1) as to fundamental frequencies, we found strong LFP responses to both low frequency (*F1* = 23Hz) and high frequency (*F2* = 200Hz), with strong main effect and some interaction (Figs 1 and 2), 2) as to harmonic and intermodulation frequencies, we found strong evidence of nonlinear processing in the somatosensory pathway (Figs 1 and 2) which were further analyzed with our computational modeling (Figs 9–12), 3) as to the spatial properties of these response, we found that they were highly spatially localized, often showing sharp response boundaries in cortical locations and isolated areas of strong responses (Figs 3–5), 4) as to the nontagged broadband (50-150Hz) high gamma power (HGP), we found evidence to question the validity of the assumption that is used to compute signal-to-noise ratio in SSEP paradigm [3, 4] observed in S2 (Figs 6–8). In the following, we will discuss each of these points in detail.

### Responses at fundamental frequencies (*f1* = 23Hz and *f2* = 200Hz)

We observed strong main effects and interactions of input stimulus amplitudes (Fig 2) at both low and high frequencies. In terms of the proportions of the recorded sites, roughly 4% and 12% of sites responded at 23Hz and 200Hz.

The dominance of the latter response type may be somewhat surprising. Previously, this type of highly narrow band oscillatory responses in LFP have been reported by Rager and Singer [31], who recorded from the visual cortex of anesthetized cats under the SSEP paradigm. However, they reported the LFP responses up to 100Hz, in response to 50Hz visual flickers. Because single neurons have refractory periods, cortical excitatory neurons usually cannot fire at 200Hz. Indeed, in our previous study [16], we did not observe any neurons that fired at that level of high frequency range. Thus, we believe that this 200Hz response is highly unlikely to originate from SUA.

There are at least two possible sources for this 200Hz LFP response, each of which is not mutually exclusive to the other. First is that it reflects high frequency inputs from other areas. LFP is known to reflect synaptic inputs to neurons rather than spiking outputs from neurons [32, 33]. Thus, if the thalamic input to these regions contain a 200Hz component, then the cortical LFP can reflect such high frequency input. Second possibility is that a population of neurons within S1 and S2 spikes in phase at 200Hz modulation [34]. In other words, while each neuron does not spike at 200Hz, population spiking is observed at 200Hz. 200Hz inputs from other areas are likely to induce subthreshold oscillation, which modulates the probability of spikes occurring in phase. This can also generate 200Hz LFP responses. As our study is not primarily designed to test these possibilities, it is not possible to draw a firm conclusion as to whether these two possibilities are sufficient (or other mechanisms play a significant role) or which possibility is dominant. Future studies are needed.

Another feature of LFP responses at fundamental frequencies is the interaction effects of both *F1* and *F2* input stimulus amplitude, which is significant at ~5% of the recorded sites (Fig 2C at both 23Hz and 200Hz responses). Clear examples of this response at single recording sites are demonstrated in Fig 5A. The interaction effects are visible in some recording sites. These interaction effects were also reported in our previous paper at the SUA level (Fig 5C–5H of [16]). For example, some SUA exhibited linear increase as the amplitude of *F1* = 23Hz increases, when *F2* = 200Hz amplitude is 0, while the same SUA did not vary at all when *F2* = 200Hz amplitude was varied, in the absence of *F1* = 23Hz modulation. However, this SUA strongly increased its response in the presence of *F2* = 200Hz. Such an interaction effect of SUAs can be explained by subthreshold oscillation and nonlinear thresholding. It is plausible that *F2* = 200Hz does not generate sufficiently strong input to a neuron to make it fire but the stimulus induces strong subthreshold oscillation, which can modulate the firing rate. This is consistent with cortical single unit analysis suggesting that high frequency drive from Pacinian corpuscle afferents has a net balanced excitatory-inhibitory drive but accounts for high frequency fluctuations in the cortical responses of many neurons [35]. Yet another possible explanation can be offered based on our LFP findings. Assuming that LFP already reflects the input from the lower thalamic areas, the above subthreshold-interaction effects are already at play in the thalamus and the interaction effects seen at S1 and S2 is just reflection of this interaction effects of thalamic neurons' spikes. Again, our experiment was not designed to test this hypothesis, thus it requires further studies to test this idea.

It should be noted that Pacinian afferents to S1 have been reported to suppress RA1-evoked activation [36, 37], but we did not observe such effects clearly (See S1–S3 Figs), which is consistent with our previous study on multi-unit activity [16]. The difference might be due to the difference in data acquisition. While we directly measured electrical activity, Tommerdahl et al. indirectly measured activity as change in oxygenation level of the cortex. Also, we used a shorter stimulus duration than those studies and results in our study reflected activities within a shorter time from the onset of stimulus.

## Evidence for nonlinear processing of somatosensory pathway

We observed strong harmonic and intermodulation (IMs) responses in both S1 and S2 across various response frequencies. While harmonic responses in LFPs have been reported previously [31], we are not aware of the reports on IM responses in LFP (for electrocorticogram, see [38]. For review of intermodulations, see [15]).

Harmonic responses at LFPs can result from purely sinusoidal spiking input modulation to a given neuron, if the synaptic input response (either excitatory postsynaptic potential or inhibitory postsynaptic potential) have non-sinusoidal form. Thus, it is not extremely informative in inferring the local neural circuit property.

On the other hand, the presence or absence of IMs are potentially quite useful. EEG studies combined with computational modeling [2, 39–41] have used IMs to distinguish potential architecture of the neural computation. In our recording at the level of LFPs, the IMs responses are often observed as indicated by blue lines along the *x*-axis in Fig 2.

Building on this empirical finding, we proceeded with the computational modeling studies (Figs 9–12). We tried to infer potential neural processes for the observed nonlinear responses by comparing some models with respect to their fitting performance. Based on the past literature, here we focused on the rectification and half squaring as the primary mechanisms, which are biologically plausible [15, 28]. Although squaring ($X^2$) and full-wave rectification (abs($X$)) are also biologically plausible, these operations do not generate responses at fundamental frequencies as they "double" inputs' frequencies (S10 Fig). While the combination of these and other operations (rectification or half squaring See Fig 10) can explain our results, we will not pursue them to restrict the possible model spaces. Thus, we did not consider these operations for modeling analysis. Overall, the quality of model fitting was similar between the rectification class and the half squaring class (Figs 11 and 12 and S8 and S9 Figs).

Further, integrative nonlinear computations were strongly implicated in our modelling in particular for S2 (Figs 11 and 12). A possible implementation of rectification at the neural circuits could be the one of phase insensitive mechanisms, such as complex cells found in the visual cortex [42]. We are not aware of specific models in the somatosensory system which would correspond to complex cells in the visual system. Meanwhile, the multiplication of two rectification functions can be implemented by coincidence-detector mechanisms, which has been found in the barn owl sound-localization system [43]. Coincidence detectors would integrate the inputs from two functions and become active in an *AND* gate like operation in a biologically plausible way. Again, we are not aware of direct evidence for coincidence-detectors in the somatosensory system.

Regardless of the precise mechanisms, our modelling analysis implicates that such mechanisms are likely to be primarily at play in S2, but not S1. While integration of the neural activity has been hypothesized to occur primarily in S2 than S1 on an anatomical basis, we believe our demonstration is the first to claim this on a computational basis. While highly intriguing, further works are needed to examine what actual neural mechanisms generate these responses in order to evaluate the usefulness of our computational modeling with IM components [15].

In this study, we investigated across-submodality interactions by using a pair of stimulus vibration frequencies (23Hz and 200Hz) which stimulates two different sensory endings: rapidly-adapting sensory endings (RAI) and very fast-adapting (PC or RAII). It is also possible to investigate within-submodality interaction by using a pair of frequencies which stimulates the same sensory endings (e.g. stimulate RAI by 25Hz and 40Hz), which might result in intermodulation even at the level before the cortical processing, resulting in different power spectrum patterns. If close tagging frequencies were used, some nonlinearities may originate at subcortical levels. Indeed, some harmonic responses were reported at the median nerve in humans

while presenting vibration stimulus [44]. Homologous findings can be found for the intermodulations as well.

## Local nature of SSEP responses

Steady state responses are most often utilized in cognitive neuroscience, typically with electroencephalography (EEG) or magnetoencephalography (MEG) performed on humans [3, 4] (but also see [1]), and predominantly using visual stimuli at relatively low frequency range (but also see [45]).

While there are some attempts to localize the source of SSEPs, due to the limitation of scalp recording, the SSEP's resolutions are still at the level of >cm. Compared to such coarse resolution, the spatial resolution of our findings is quite striking. For example, Fig 2 shows that not all channels reflected SSEPs, suggesting at least the equivalent level of spatial resolution is necessary for localizing their sources. (See Figs 3–5 and 7 as well). Our spatial resolution was potentially enhanced by bipolar re-referencing [17, 46] and may reflect highly localized clusters of neurons or synaptic inputs to such neurons. The isolated response patterns in Figs 3–5 and 7 suggest that such localized circuitry may exist in a submillimeter scale, the spacing of our array electrodes. Combining with a dense EEG/MEG/ECoG recording, the usage of SSEP paradigm may be able to push the limit of spatial resolution of the source localization for these techniques, by targeting highly localized neural circuits within the human brain. Given the recent explosion of brain machine interface applications that utilize the SSEP paradigm [3], an improved spatial resolution of SSEP can have broader implications outside of the basic science, extending to the field of engineering and clinical neuroscience.

## High gamma power (HGP) responses in S2 and log*SNR* computation

Outside of the tagged frequencies (fundamental, harmonics, and IMs), we initially expected that LFP power spectra can be regarded as noise, as is usually assumed in SSEP literature [3, 4]. However, we found that this assumption was not really met in some channels (~5%) within S2 (Fig 6). One might consider the observed broadband (50-150Hz) HGP as aliasing artifacts that band-pass filtering produces in its lower frequency range. We believe the observed HGP is unlikely to be due to the artifacts for four reasons. First, we did not perform any band-pass filtering to the recording in S1 and S2. Second, HGP were observed only in S2 not in S1 even though we performed the same analysis for S1 and S2 recordings. Third, we did not observe strong responses at any other multiples of the HGP range even up to 5000Hz (S11 Fig). Fourth, such broadband HGP around 50-150Hz has been reported in some studies that analyzed ECoG [47–49] and LFPs [50].

Having said that, the HGP responses documented in these studies tend to be short-lived and to disappear soon after the stimulus onset. Thus, the evoked HGP may not be relevant for log*SNR* computation if only the sustained components are used. However, our S2 HGP were in fact sustained and at the comparable magnitude with those at the harmonic and intermodulation frequencies (Fig 8).

We are not sure why HGP was completely absent in S1. Along with the above modeling results, the presence or absence of HGP was a prominent difference between S1 and S2. This difference in HGP might be partially explained by the difference in recording electrodes (planar array for S1 and linear array for S2).

While the exact reason of the presence of HGP in S2 but not in S1 remains puzzling, our results at least provide a cautionary note on the practice of log*SNR* computation, which assumes that evoked SSEP responses can be quantified by comparisons of the power at a given

stimulation frequency f and its neighbors. Neighboring non-stimulation frequencies can indeed change its response magnitude in the SSEP paradigm.

## Conclusion

Taken together, our SSEP analyses with LFP revealed nonlinear processing in somatosensory cortex, which is likely to be hierarchically organized. Our results constrain the computational and hierarchical organization of S1 and S2, implying S2 integrates oscillatory functions in a nonlinear way. While this paper focused on a simple Fourier transform of the LFPs, interactions among simultaneously recorded LFPs can be further analyzed by other types of spectral-domain techniques, ranging from coherency [51], Granger causality [52, 53], and integrated information [54]. Given the rich information present in LFPs, which reflects the local neural circuit properties, especially IMs [15], this type of high resolution LFP recordings from animal models can serve as quite powerful tools to dissect the functional and computational properties of underlying circuitries of neural systems across sensory modalities.

## Supporting information

**S1 Fig. log*SNR* as functions of stimulus amplitudes (# of conditions = 6).** (A) log*SNR* at *f1* = 23Hz as functions of *F1* stimulus amplitudes for S1 and S2. We computed the mean and standard error across trials per bipolar channel in Session 1–1 and 1–2, which had 6 amplitude conditions for each *F1* and *F2* vibration stimulus. The point in the graph is the mean across the channels and the error bar is the mean of the standard errors across the channels. Colour encodes *F2* stimulus amplitudes. (B) log*SNR* at *f2* = 200Hz as functions of *F2* stimulus amplitudes for S1 and S2. Colour encodes *F1* stimulus amplitudes.
(TIF)

**S2 Fig. log*SNR* as functions of stimulus amplitudes (# of conditions = 5).** (A) log*SNR* at *f1* = 23Hz as functions of *F1* stimulus amplitudes for S1 and S2. The same format as S1 Fig. We included Session 1–3, 2–4, 2–5 and 2–6, which had 5 amplitude conditions. Note that Session 1–3 had different *F2* stimulus amplitudes from the other sessions, these values that are shown in brackets. (B) log*SNR* at *f2* = 200Hz as functions of *F2* stimulus amplitudes for S1 and S2.
(TIF)

**S3 Fig. logSNR as functions of stimulus amplitudes (# of condition = 4).** (A) logSNR at f1 = 23Hz as functions of F1 stimulus amplitudes for S1 and S2. The same format as S1 Fig. We included Session 2–1, 2–2 and 2–3, which had 4 amplitude. (B) logSNR at f2 = 200Hz as functions of F2 stimulus amplitudes for S1 and S2.
(TIF)

**S4 Fig. Exemplar *VELogP* at *f1* = 23Hz depend on the vibration amplitude of *F1* = 23Hz.** Shown with the same format as Fig 3. (A) *VELogP* of bipolar channel 176 in S1 (Session 2–2). This channel's responses at *f1* = 23Hz showed a significant main effect of *F1* = 23Hz amplitude only. *p*-value (*F1*, *F2*, interaction) = ($<10^{-5}$, 0.20, 0.042) with the corrected threshold 0.00016. (B) Spatial mapping of *VELogP* at *f1* = 23Hz across all channels in S1 (Session 2–2).
(TIF)

**S5 Fig. Exemplar *VELogP* at *f2* = 200Hz depend on the vibration amplitude of *F2* = 200Hz.** Shown with the same format as Fig 3. (A) *VELogP* of bipolar channel 122 in S1 (Session 2–2). This channel's responses at *f2* = 200Hz showed a significant main effect of *F2* = 200Hz amplitude only. *p*-value (*F1*, *F2*, interaction) = (0.0033, $<10^{-5}$, 0.045) with the corrected threshold

0.00016. (B) Spatial mapping of *VELogP* at *f2* = 200Hz across all channels in S1 (Session 2–2).
(TIF)

**S6 Fig. Exemplar *VELogP* at *f1* = 23Hz depend on the vibration amplitude of *F1* = 23Hz, *F2* = 200Hz, and their interaction.** Shown with the same format as Fig 3. (A) *VELogP* of channel 43 in S1 (Session 2–2). This channel's responses at *f1* = 23Hz showed a significant main effect of *F1* = 23Hz amplitude, the main effect of *F2* = 200Hz, and their interaction. *p*-values (*F1*, *F2*, interaction) are all $p < 10^{-5}$ with the corrected threshold 0.00016. (B) Spatial mapping of *VELogP* at *f1* = 23Hz across all channels in S1 (Session 2–2).
(TIF)

**S7 Fig. Exemplar channel's observed and modelled responses for half squaring models (HSq).** (A) Bipolar channel 43 in S1 (Session 2–2)'s log*S*NR (black) is compared to the optimally fitted models from four half squaring models. Adding *HSq*(*XY*) to two models (HSq-1 and HSq-3 in Fig 12) did not change the results and we showed four models in two colors: red for HSq-1 & HSq-2 and blue for HSq-3 & HSq-4. The best parameters for each model are the following. HSq-1 (red): -1.0*HSq*(*X*)+64*HSq*(*Y*), HSq-2 (red): -0.54*HSq*(*X*)+34*HSq*(*Y*) $-1.4^{*}10^{308}HSq(XY)$, HSq-3 (blue): -0.94*HSq*(*X*)+38*HSq*(*Y*)+1.4*HSq*(*X*)*HSq*(*Y*) and HSq-4 (blue): -0.85*HSq*(*X*)+35*HSq*(*Y*)$-1.4^{*}10^{308}HSq(XY)$+1.3*HSq*(*X*)*HSq*(*Y*). The respective sums of log*S*NR differences from the observed log*S*NR across the frequencies of interest are 125 and 74. (B) Difference between the observed and the best model at the frequencies of interest. Color scheme is the same as in (A).
(TIF)

**S8 Fig. Distribution of each coefficient in rectification models.** Cumulative probability distribution of each coefficient in S1 (A) and S2 (B). For a display purpose, we removed the top 3% and bottom 3% channels. This summarizes actual values of each parameter in rectification models. It also shows how adding a nonlinear component had an effect on other parameters. e.g. the top left panel in A shows that the model with *Rect*(*X*) (purple line) and the full model (yellow line) seemed to have similar the parameter a (i.e. coefficient of *Rect*(*X*) overall. On the other hand, the other two models (blue and red) seemed to have different parameters overall.
(TIF)

**S9 Fig. Comparison of fitting performances across half squaring models.** (A) Comparison of performance across four half squaring models based on the cumulative probability distributions of the minimum difference for S1 and S2 separately. Adding *HSq*(*XY*) did not change the results and we showed four models in two colors: red for HSq-1 and HSq-2 and blue for HSq-3 and HSq-4. (B) Comparison of each model's performance between S1 and S2 based on the cumulative probability distributions of the minimum difference. We showed the two models giving the same distribution curves as one subplot. *** indicates p<0.001 according to Kolmo-gorov-Smirnov tests.
(TIF)

**S10 Fig. Waveforms of various modelled response types.** (A) Waveforms in the time domain (0 to 0.25s). *X* and *Y* are sinusoidal inputs at 23Hz and 200Hz, respectively. *Pow2* represents the squaring (i.e. $X^2$) and *FWRect* represents the full-wave rectification. Note that, for these operations, multiplication inside and outside of functions are identical with each other i.e. $f(X)^{*}f(Y) = f(X^{*}Y)$. (B) Spectra of each waveform in the frequency domain. All of these models do not generate responses at fundamental frequencies. (C) Noise can suppress responses at harmonic frequencies in *Rect(X)*. Waveforms in the time domain (0 to 0.1s). Noise is given by normal distribution $N(0, 0.5^2)$ (as in the simulation reported in Gordon et al. 2019). (D)

Spectra of each waveform in the frequency domain. Adding noise to the input suppresses responses at some harmonic frequencies.
(TIF)

**S11 Fig.** VELogP from top 10% channels in S2 across A) 0-5000Hz and B) 0-1000Hz). The top 10% channels in S2 showing higher HGP responses. We took the mean and standard error across trials for each channel and then took their respective means across channels. Red vertical lines show 50Hz and 150Hz. For a display purpose, we removed responses at fundamentals, harmonics and some intermodulation frequencies.
(TIF)

**S1 Appendix. Line search optimisation technique.**
(PDF)

**S2 Appendix. Akaike's information criterion.**
(PDF)

## Acknowledgments

We thank Prof Trichur Vidyasagar for his insightful comments.

## Author Contributions

**Conceptualization:** Spencer Chin-Yu Chen, Richard Martin Vickery, John W. Morley, Naotsugu Tsuchiya.

**Formal analysis:** Yota Kawashima, Rannee Li, Naotsugu Tsuchiya.

**Funding acquisition:** John W. Morley, Naotsugu Tsuchiya.

**Investigation:** Spencer Chin-Yu Chen, Richard Martin Vickery, John W. Morley.

**Methodology:** Yota Kawashima, Rannee Li, Naotsugu Tsuchiya.

**Project administration:** Spencer Chin-Yu Chen, Richard Martin Vickery, John W. Morley, Naotsugu Tsuchiya.

**Resources:** Spencer Chin-Yu Chen, Richard Martin Vickery, John W. Morley, Naotsugu Tsuchiya.

**Software:** Yota Kawashima, Rannee Li, Naotsugu Tsuchiya.

**Supervision:** Naotsugu Tsuchiya.

**Visualization:** Yota Kawashima, Rannee Li, Naotsugu Tsuchiya.

**Writing – original draft:** Yota Kawashima, Rannee Li, Spencer Chin-Yu Chen, Naotsugu Tsuchiya.

**Writing – review & editing:** Yota Kawashima, Rannee Li, Spencer Chin-Yu Chen, Richard Martin Vickery, John W. Morley, Naotsugu Tsuchiya.

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
