## [Decision Letter · Decision Letter 0]

8 Dec 2020

PONE-D-20-29400

Steady state evoked potential (SSEP) responses in the primary and secondary somatosensory cortices of anesthetized cats: nonlinearity characterized by harmonic and intermodulation frequencies

PLOS ONE

Dear Dr. Tsuchiya,

Thank you for submitting your manuscript to PLOS ONE and apologies for the slow processing of it. We had problems with finding appropriate reviewers and receive reviews on time. 

After careful consideration, we feel that it has merit but does not fully meet PLOS ONE’s publication criteria as it currently stands. Therefore, we invite you to submit a revised version of the manuscript that addresses the points raised during the review process.

The reviewers find the research topic important and the manuscript mostly well written. 

They both bring up a range of questions regarding the experiments and the data sets. The manuscript could be much clearer here and the authors must attempt to answer the questions in sufficient depth. 

Both authors have further, minor comments that the authors should consider in order to improve the manuscript. 

Please also note that PLOS ONE expects that data sets underlying published manuscripts are properly published, either in the main text, as supplementary material, or in a publicly accessible repository. 

We look forward to receiving your revised manuscript.

Kind regards,

Thomas Wennekers

Academic Editor

PLOS ONE

Journal Requirements:

2. To comply with PLOS ONE submissions requirements, in your Methods section, please provide additional information on the animal research and ensure you have included details on (1) methods of sacrifice, (2) the source of the animals, and (3) details of housing and care.

"This work was supported by the MASSIVE HPC facility (www.massive.org.au)."

"NT was supported by Australian Research Council (funding number: FT120100619, DP130100194, DP180104128, and DP180100396).

URL: https://www.arc.gov.au/

JM was supported by Australian Research Council Thinking Systems Grant TS0669860. URL: " ext-link-type="uri" xlink:type="simple">https://www.arc.gov.au/"

Reviewers' comments:

Reviewer's Responses to Questions

**Comments to the Author**

1. Is the manuscript technically sound, and do the data support the conclusions?

Reviewer #1: Yes

Reviewer #2: Yes

2. Has the statistical analysis been performed appropriately and rigorously? 

Reviewer #1: Yes

Reviewer #2: Yes

3. Have the authors made all data underlying the findings in their manuscript fully available?

Reviewer #1: No

Reviewer #2: Yes

4. Is the manuscript presented in an intelligible fashion and written in standard English?

Reviewer #1: Yes

Reviewer #2: Yes

5. Review Comments to the Author

Reviewer #1: Review of Kawashima et al., “Steady state evoked potential responses in the primary and secondary somatosensory cortices of anaesthetized cats”, submitted to PLoS ONE.

Summary

This study reports local field potential recordings from cat somatosensory cortex, during vibrotactile stimulation of the paw. This is a very interesting topic, using a technique that is widespread in studying other senses, but which has not yet been fully exploited for somatosensation. The data appear to be of good quality, though I have some queries about the choice of frequencies, and also suggestions about the way the results are presented. There are also some minor comments listed below that should not be difficult to fix or clarify.

Specific points

1. This is an excellent data set. But I feel that it could be presented with much greater clarity. For example, I did not understand the purpose of plotting the % of significant channels (Figure 2). As an alternative, why not plot the signal strength (either SNR, power or VELogP, I don’t have a particular preference) as a function of the stimulus amplitude? That would allow you to show modulation-response functions. For example, you could plot the amplitude of the 23Hz stimulus on the x-axis and the response at 23Hz on the y-axis, with different functions for different amplitudes of 200Hz stimulation. Then, if the 200Hz component affects the 23Hz response, you will see a ‘gain control’ type effect. This is the standard way of looking at such data for other senses such as vision (see e.g. Busse et al,. 2009, Neuron; Carandini Heeger, 2012, Nat Rev Neurosci; Tsai et al., 2012, J Neurosci), and would allow for more straightforward comparisons across studies, as well as being generally clearer.

2. I thought the attempt to model the pattern of intermodulation responses was impressive. However I have not come across the ‘square wave’ nonlinearity before. Is this something that neurons have plausibly been shown to implement? I would have thought a squaring (or other power law) nonlinearity would be a more obvious choice to investigate, or even full-wave (for comparison with half-wave) rectification.

3. I was surprised that the stimulation frequencies were so different from each other, and that 200Hz was chosen. Would the authors expect that the same neurons driven by 23Hz stimulation could also respond to 200Hz stimulation, or would these be likely to be distinct populations? Would you expect very different results if you used more similar frequencies that would be more likely to fall in a single neuron’s frequency response, such as 23Hz and 27Hz, or 200Hz and 187Hz? Why was 200Hz chosen in the first place, given that cortical neurons typically do not respond this fast (as the authors point out in the Discussion)?

4. Typically, frequencies beyond the pass-band of a filter can produce lower frequency aliasing artifacts. Might the broadband response be explained by aliasing artifacts from a population of neurons that all have slightly different temporal properties, being driven by a high frequency beyond their usual range (e.g. 200Hz)?

5. The data availability statement says that the data will be made available after acceptance of the manuscript. I believe that assessing data quality and availability is an important part of peer review, and I think that the data should at least be made available to the reviewers upon resubmission (OSF permits this with ‘read only’ links).

6. Line 149 (and elsewhere), I didn’t understand the need for inverted commas around “deviation”.

7. Conversions to power and logSNR are described clearly. But in the figures, logSNR is expressed in dB. Conventionally this involves multiplying the raw log values by some factor (usually 10 or 20), but I couldn’t see this specified anywhere.

8. The rationale for the analysis given in Figure 2 was rather unclear. Why should we care about the % of channels that show one or other response, or their interaction? It wasn’t obvious to me what this was telling us.

9. If each trial is 2 seconds long, the spectral resolution should be 1/2 = 0.5Hz, as stated on line 210. But it appears to be much higher than this in Figure 3A. Why?

10. The top left panel of Fig7a (and Supplementary Figs 1-3) shows a completely flat line, but with grey error regions similar to the other panels. Is this a plotting error? It looks fine in Figs 3-5.

11. Very hard to see what’s happening clearly in Figure 10a. I’d suggest offsetting each trace vertically by some fixed amount, so that we can see all of them clearly.

12. I would like to see a clear description of what the cumulative probability curves in Figure 11 (and Supplementary Figure 5 6) represent.

Reviewer #2: The manuscript by Tsuchiya et al. explores nonlinear nature of mechanoreceptor input transformation in cat’s S1 and S2. The basic methodological approach they use is straightforward: if a dynamical system is strictly linear, its output in response to a stable sinusoidal input signal will be a signal of the exactly the same frequency and varying only in its amplitude and phase shift. In the frequency domain, the power spectrum of such output signal will have only a single spike, located at the input frequency. Furthermore, if a linear system is driven by a sum of 2 or more sinusoidal inputs of different frequencies, the output power spectrum will be a sum of power spectra of constituent frequencies. In contrast, if a dynamical system is nonlinear, there will be additional spikes in the power spectrum at various harmonics of the input frequency, at intermodulation frequencies, etc.

Approaching somatosensory system as a dynamical system in this theoretical framework, the authors interrogated it by (1) driving it with peripheral sinusoidal inputs; (2) using LFPs, recorded at various locations in S1 and S2, as the system’s output signals; and (3) displaying these output signals in the frequency domain. Peripheral input signals were in a form of sinusoidal vertical skin displacement oscillations applied to the central pad or the pads of D4 or D5 at a combination of 23Hz and 200Hz frequencies, using different amplitude pairings (ranging from zero to modest strength). Not surprisingly – since we already know that the nervous system is profoundly nonlinear – power spectra of cortical stimulus-evoked LFPs had plenty of spikes at harmonics and intermodulation frequencies. Different recording sites produced a variety of power spectra, some of which might be due to their location in either S1 or S2, or being in the center or margin of the responding cortical region, or particular cortical layer.

The manuscript is clearly written, data analyses and figures are informative and, overall, the paper offers an effective demonstration of the usefulness of this steady-state frequency-domain technique for interrogating somatosensory cortical networks and computational operations they perform on their afferent inputs (which can be used in practical experimental validation of cortical models).

Specific Comments

(1) Some of nonlinearities observed in S1 and S2 power spectra might originate at subcortical levels. For example, Kelly et al. (Exp. Brain Res. 109:500-506, 1996) recorded activity of 1st order afferents in human median nerve while delivering flutter stimulation to D2 at 33Hz. Their power spectrum shows a clear 1st harmonic (see their Fig. 1) only about 50% smaller (on log10 scale) than at the driving frequency. This power spectrum is very similar to what you get after rectifying a cosine function.

(2) Lines 541-542, also Fig. 9: The text states: “As is reviewed by Gordon et al. 2019, these nonlinear operations, Rect(X) and Sq(X), alone do not generate responses at harmonic nor intermodulation frequencies (Fig 9)”. Actually, Rect(X) does generate a substantial response at the 1st harmonic. Gordon et al. (2019) example in fact shows that harmonic clearly. Fig. 9B for Rect(X) in this manuscript shows spikes of equal magnitudes at all odd harmonics, which is not right.

(3) The two stimulus frequencies used in this study (23Hz flutter and 200Hz vibration) preferentially engage different afferent submodalities (RA1 and Pacinian), which have different projections to somatosensory cortex. Power spectrum patterns might change significantly, compared to the ones presented in this paper, if the two stimulus frequencies are confined to the same submodality (e.g., 25Hz and 40Hz). Within-submodality interactions in S1 and S2 cortical columns might be different from across-submodality interactions, and this frequency-domain technique might be able to demonstrate this difference very nicely.

(4) According to Abstract, the cats were studied under ketamine-xylazine general anesthesia, but on page 4 (line 127) it is stated that anesthesia was maintained with alfaxalone. Which one was it?

(5) According to Whitsel and colleagues (Tommerdahl et al. J. Neurophysiol. 82:16-33, 1999; Tommerdahl et al., Somatosens. Mot. Res. 22:151-169, 2005), Pacinian projections to S1 have prominent suppressive effects on RA1-evoked activation there. No such suppression of the 23Hz stimulus by the 200Hz stimulus is evident in this paper. Some discussion of this disagreement would be desirable. The only significant difference that I can see between the experimental conditions is the anesthetic used: Whitsel et al. used 50/50 nitrous oxide in oxygen (and neuromuscular block) for a very light level of anesthesia, whereas this study used either ketamine or alfaxalone. If it was ketamine, then it might be responsible, since ketamine greatly reduces stimulus-evoked dynamics in S1.

(6) Line 299: change “LEP” to “LFP”

(7) Line 587: “30Rect(X)” is probably “30Rect(Y)”

(8) Line 613: change “Smironv” to “Smirnov”

(9) Line 855: reference #20 uses first names of the authors rather than their surnames. So, Gyorgy, Costas, Christof should be changed to Buzsaki, Anastassiow, Koch.

6. PLOS authors have the option to publish the peer review history of their article (what does this mean?). If published, this will include your full peer review and any attached files.

Reviewer #1: No

Reviewer #2: **Yes: **Oleg V. Favorov

---

## [Author Response · Author response to Decision Letter 0]

10 Jan 2021

Dear Dr. Thomas Wennekers,

We sincerely appreciate your time for evaluating our manuscript and considering it for publication in PLOS ONE.

We carefully considered and addressed all comments from the reviewers. Specifically: 

1. We clarified various aspects of the data and experiments to address reviewers’ comments. 

2. We addressed all minor comments from the reviewers. 

3. We have made the data publicly available online at OSF (DOI: 10.17605/OSF.IO/CQRFJ)

We also added details on (1) methods of sacrifice, (2) the source of the animals, and (3) details of housing and care in the Methods section.

Regarding the Acknowledgements, the information we provided (the MASSIVE HPC facility) is not a funding source. Therefore, we keep the Acknowledgements as it is.

With these revisions, we believe that our revised manuscript has improved significantly and addressed all the comments. Please also see our point-by-point responses to the reviewers’ comments in the submitted file (Respond to Reviewers.pdf). We believe that the manuscript is now clear and suitable for publication in PLOS ONE.

Regards,

Naotsugu Tsuchiya (on behalf of the coauthors)

---

## [Decision Letter · Decision Letter 1]

11 Feb 2021

Steady state evoked potential (SSEP) responses in the primary and secondary somatosensory cortices of anesthetized cats: nonlinearity characterized by harmonic and intermodulation frequencies

PONE-D-20-29400R1

Dear Dr. Tsuchiya,

We’re pleased to inform you that your manuscript has been judged scientifically suitable for publication and will be formally accepted for publication once it meets all outstanding technical requirements.

Kind regards,

Thomas Wennekers

Academic Editor

PLOS ONE

Reviewers' comments:

Reviewer's Responses to Questions

**Comments to the Author**

1. If the authors have adequately addressed your comments raised in a previous round of review and you feel that this manuscript is now acceptable for publication, you may indicate that here to bypass the “Comments to the Author” section, enter your conflict of interest statement in the “Confidential to Editor” section, and submit your "Accept" recommendation.

Reviewer #1: All comments have been addressed

2. Is the manuscript technically sound, and do the data support the conclusions?

Reviewer #1: Yes

3. Has the statistical analysis been performed appropriately and rigorously? 

Reviewer #1: Yes

4. Have the authors made all data underlying the findings in their manuscript fully available?

Reviewer #1: Yes

5. Is the manuscript presented in an intelligible fashion and written in standard English?

Reviewer #1: Yes

6. Review Comments to the Author

Reviewer #1: (No Response)

7. PLOS authors have the option to publish the peer review history of their article (what does this mean?). If published, this will include your full peer review and any attached files.

Reviewer #1: No

---

## [Editor Report · Acceptance letter]

26 Feb 2021

PONE-D-20-29400R1 

Steady state evoked potential (SSEP) responses in the primary and secondary somatosensory cortices of anesthetized cats: nonlinearity characterized by harmonic and intermodulation frequencies 

Dear Dr. Tsuchiya:

I'm pleased to inform you that your manuscript has been deemed suitable for publication in PLOS ONE. Congratulations! Your manuscript is now with our production department. 

Kind regards, 

on behalf of

Dr. Thomas Wennekers 

Academic Editor

PLOS ONE